# Window-Based Distribution Shift Detection for Deep Neural Networks

**Guy Bar-Shalom**
Department of Computer Science
Technion - Israel Institute of Technology
guy.b@cs.technion.ac.il

**Yonatan Geifman**
Deci.AI
yonatan@deci.ai

**Ran El-Yaniv**
Department of Computer Science
Technion - Israel Institute of Technology, Deci.AI
rani@cs.technion.ac.il

## Abstract

To deploy and operate deep neural models in production, the quality of their predictions, which might be contaminated benignly or manipulated maliciously by input distributional deviations, must be monitored and assessed. Specifically, we study the case of monitoring the healthy operation of a deep neural network (DNN) receiving a stream of data, with the aim of detecting input distributional deviations over which the quality of the network's predictions is potentially damaged. Using selective prediction principles, we propose a distribution deviation detection method for DNNs. The proposed method is derived from a tight coverage generalization bound computed over a sample of instances drawn from the true underlying distribution. Based on this bound, our detector continuously monitors the operation of the network over a test window and fires off an alarm whenever a deviation is detected. Our novel detection method performs on-par or better than the state-of-the-art, while consuming substantially lower computation time (five orders of magnitude reduction) and space complexity. Unlike previous methods, which require at least linear dependence on the size of the source distribution for each detection, rendering them inapplicable to "Google-Scale" datasets, our approach eliminates this dependence, making it suitable for real-world applications. Code is available at https://github.com/BarSGuy/Window-Based-Distribution-Shift-Detection.

## 1 Introduction

A wide range of artificial intelligence applications and services rely on deep neural models because of their remarkable accuracy. When a trained model is deployed in production, its operation should be monitored for abnormal behavior, and a flag should be raised if such is detected. Corrective measures can be taken if the underlying cause of the abnormal behavior is identified. For example, simple distributional changes may only require retraining with fresh data, while more severe cases may require redesigning the model (e.g., when new classes emerge).

In this paper we focus on distribution shift detection in the context of deep neural models and consider the following setting. Pretrained model $f$ is given, and we presume it was trained with data sampled from some distribution $P$. In addition to the dataset used in training $f$, we are also given an additional unlabeled sample of data from the same distribution, which is used to train a detector $D$ (we refer to this as the detection-training set or source set). While $f$ is used in production to process a stream of emerging input data, we continually feed $D$ with the most recent window $W_k$ of $k$ input elements.

37th Conference on Neural Information Processing Systems (NeurIPS 2023).

The detector also has access to the final layers of the model $f$ and should be able to determine whether the data contained in $W_k$ came from a distribution different from $P$. We emphasize that in this paper we are not considering the problem of identifying *single-instance* out-of-distribution or outlier instances [28, 21, 22, 16, 38, 35, 34, 11], but rather the information residing in a population of $k$ instances. While it may seem straightforward to apply single-instance detectors to a window (by applying the detector to each instance in the window), this approach can be computationally expensive since such methods are not designed for window-based tasks; see discussion in Section 3. Moreover, we demonstrate here that *computationally feasible* single-instance methods can fail to detect population-based deviations. We emphasize that we are not concerned in characterizing types of distribution shifts, nor do we tackle at all the complementary topic of out-of-distribution robustness.

Distribution shift detection has been scarcely considered in the context of deep neural networks (DNNs), however, it is a fundamental topic in machine learning and statistics. The standard method for tackling it is by performing a dimensionality reduction over both the detection-training (source) and test (target) samples, and then applying a two-sample statistical test over these reduced representations to detect a deviation. This is further discussed in Section 3. In particular, deep models can benefit from the semantic representation created by the model itself, which provides meaningful dimensionality reduction, that is readily available at the last layers of the model. Using the embedding layer (or softmax) along with statistical two-sample tests was recently proposed by Lipton et al. [29] and Rabanser et al. [37] who termed solutions of this structure black-box shift detection (BBSD). Using both the univariate Kolmogorov-Smirnov (KS) test and the maximum mean discrepancy (MMD) method, see details below, Rabanser et al. [37] achieve impressive detection results when using MNIST and CIFAR-10 as proxies for the distribution $\mathcal{P}$. As shown here, the KS test is also very effective over ImageNet when a stronger model is used (ResNet50 vs ResNet18). However, BBSD methods have the disadvantage of being computationally intensive (and probably inapplicable to read-world datasets) due to the use of two-sample tests between the detection-training set (which can, and is preferred to be the largest possible) and the window $W_k$; a complexity analysis is provided in Section 5.

We propose a different approach termed *Coverage-Based Detection* (CBD), which builds upon selective prediction principles [9, 14]. In this approach, a model quantifies its prediction uncertainty and abstains from predicting uncertain instances. First, we develop a method for selective prediction with guaranteed coverage. This method identifies the best abstaining threshold and coverage bound for a given pretrained classifier $f$, such that the resulting empirical coverage will not violate the bound with high probability (when abstention is determined using the threshold). The guaranteed coverage method is of independent interest, and it is analogous to selective prediction with guaranteed risk [14]. Because the empirical coverage of such a classifier is highly unlikely to violate the bound if the underlying distribution remains the same, a systematic violation indicates a distribution shift. To be more specific, given a detection-training sample $S_m$, our coverage-based detection algorithm computes a fixed number of tight generalization coverage bounds, which are then used to detect a distribution shift in a window $W_k$ of test data. The proposed detection algorithm exhibits remarkable computational efficiency due to its ability to operate independently of the size of $S_m$ during detection, which is crucial when considering "Google-Scale" datasets, such as the JFT-3B dataset. In contrast, the currently available distribution shift detectors rely on a framework that requires significantly higher computational requirements (this framework is illustrated in Figure 3 in Appendix A). A run-time comparison of those baselines w.r.t. our CBD method is provided in Figure 1.

In a comprehensive empirical study, we compared our CBD algorithm with the best-performing baselines, including the KS approach of [37]. Additionally, we investigated the suitability of single-instance detection methods for identifying distribution shifts. For a fair comparison, all methods used the same (publicly available) underlying models: ResNet50, MobileNetV3-S, and ViT-T. To evaluate the effectiveness of our approach, we employed the ImageNet dataset to simulate the source distribution. We then introduced distribution shifts using a range of methods, starting with simple noise and progressing to more sophisticated techniques such as adversarial examples. Based on these experiments, it is evident that CBD is overall significantly more powerful than the baselines across a wide range of test window sizes.

To summarize, the contributions of this paper are: (1) A theoretically justified algorithm (Algorithm 1), that produces a coverage bound, which is of independent interest, and allows for the creation of selective classifiers with guaranteed coverage. (2) A theoretically motivated "windowed" detection

algorithm, CBD (Algorithm 2), which detects a distribution shift over a window; this proposed algorithm exhibits remarkable efficiency compared to state-of-the-art methods (five orders of magnitude better than the best method). (3) A comprehensive empirical study demonstrating significant improvements relative to existing baselines, and introducing the use of single-instance methods for detecting distribution shifts.

## 2  Problem Formulation

We consider the problem of detecting distribution shifts in input streams provided to pretrained deep neural models. Let $\mathcal{P} \triangleq P_X$ denote a probability distribution over an input space $\mathcal{X}$, and assume that a model $f$ has been trained on a set of instances drawn from $\mathcal{P}$. Consider a setting where the model $f$ is deployed and while being used in production its input distribution might change or even be attacked by an adversary. Our goal is to detect such events to allow for appropriate action, e.g., retraining the model with respect to the revised distribution.

Inspired by Rabanser et al. [37], we formulate this problem as follows. We are given a pretrained model $f$, and a detection-training set, $S_m \sim \mathcal{P}^m$. Then we would like to train a detection model to be able to detect a distribution shift; namely, discriminate between windows containing in-distribution (ID) data, and *alternative-distribution* (AD) data. Thus, given an unlabeled test sample window $W_k \sim Q^k$, where $Q$ is a possibly different distribution, the objective is to determine whether $\mathcal{P} \neq Q$. We also ask what is the smallest test sample size $k$ required to determine that $\mathcal{P} \neq Q$. Since typically the detection-training set $S_m$ can be quite large, we further ask whether it is possible to devise an effective detection procedure whose time complexity is $o(m)$.

## 3  Related Work

Distribution shift detection methods often comprise the following two steps: dimensionality reduction, and a two-sample test between the detection-training sample and test samples. In most cases, these methods are "lazy" in the sense that for each test sample, they make a detection decision based on a computation over the entire detection-training sample. Their performance will be sub-optimal if only a subset of the train sample is used. Figure 3 in Appendix A illustrates this general framework.

The use of dimensionality reduction is optional. It can often improve performance by focusing on a less noisy representation of the data. Dimensionality reduction techniques include no reduction, *principal components analysis* [1], *sparse random projection* [2], *autoencoders* [39, 36], *domain classifiers*, [37] and more. In this work we focus on *black box shift detection* (BBSD) methods [29], that rely on deep neural representations of the data generated by a pretrained model. The representation extracted from the model will typically utilize either the softmax outputs, acronymed BBSD-Softmax, or the embeddings, acronymed BBSD-Embeddings; for simplicity, we may omit the BBSD acronym. Due to the dimensionality of the final representation, multivariate or multiple univariate two-sample tests can be conducted.

By combining BBSD-Softmax with a Kolmogorov-Smirnov (KS) statistical test [33] and using the Bonferroni correction [3], Rabanser et al. [37] achieved state-of-the-art results in distribution shift detection in the context of image classification (MNIST and CIFAR-10). We acronym their method as BBSD-KS-Softmax (or KS-Softmax). The *univariate* KS test processes individual dimensions separately; its statistic is calculated by computing the largest difference $Z$ of the *cumulative density functions* (CDFs) across all dimensions as follows: $Z = \sup_{\boldsymbol{z}} |F_{\mathcal{P}}(\boldsymbol{z}) - F_Q(\boldsymbol{z})|$, where $F_Q$ and $F_{\mathcal{P}}$ are the empirical CDFs of the detection-training and test data (which are sampled from $\mathcal{P}$ and $Q$, respectively; see Section 2). The Bonferroni correction rejects the null hypothesis when the minimal p-value among all tests is less than $\frac{\alpha}{d}$, where $\alpha$ is the significance level and $d$ is the number of dimensions. Although less conservative approaches to aggregation exist [20, 30], they usually assume some dependencies among the tests. The *maximum mean discrepancy* (MMD) method [18] is a kernel-based multivariate test that can be used to distinguish between probability distributions $\mathcal{P}$ and $Q$. Formally, $MMD^2(\mathcal{F}, \mathcal{P}, Q) = ||\boldsymbol{\mu}_{\mathcal{P}} - \boldsymbol{\mu}_Q||^2_{\mathcal{F}^2}$, where $\boldsymbol{\mu}_{\mathcal{P}}$ and $\boldsymbol{\mu}_Q$ are the mean embeddings of $\mathcal{P}$ and $Q$ in a reproducing kernel Hilbert space $\mathcal{F}$. Given a kernel $\mathcal{K}$, and samples, $\{x_1, x_2, \ldots, x_m\} \sim \mathcal{P}^m$ and $\{x'_1, x'_2, \ldots, x'_k\} \sim Q^k$, an unbiased estimator for $MMD^2$ can be found in [18, 42]. Sutherland et al. [43] and Gretton et al. [18] used the RBF kernel $\mathcal{K}(x, x') = e^{-\frac{1}{2\sigma^2}||x-x'||^2_2}$, where $2\sigma^2$ is set to the median of the pairwise Euclidean distances

between all samples. By performing a permutation test on the kernel matrix, the p-value is obtained. In our experiments (see Section 6.4), we thus use four population based baselines: **KS-Softmax**, **KS-Embeddings**, **MMD-Softmax**, and **MMD-Embeddings**.

As mentioned in the introduction, our work is complementary to the topic of single-instance out-of-distribution (OOD) detection [28, 21, 22, 16, 38, 35, 34, 11]. Although these methods can be applied to each instance in a window, they often fail to capture population statistics within the window, making them inadequate for detecting population-based changes. Additionally, many of these methods are computationally expensive and cannot be applied efficiently to large windows. For example, methods such as those described in [41, 27] extract values from each layer in the network, while others such as [28] require gradient calculations. We note that the application of the best single-instance methods such as [41, 27, 28] in our (large scale) empirical setting is computationally challenging and preclude empirical comparison to our method. Therefore, we consider (in Section 6.4) the detection performance of two computationally efficient single-instance baselines: Softmax-Response (abbreviated as **Single-instance SR** or Single-SR) and Entropy-based (abbreviated as **Single-instance Ent** or Single-Ent), as described in [4, 6, 21]. Specifically, we apply each single-instance OOD detector to every instance in the window and in the detection-training set. We then use a two-sample t-test to determine the p-value between the uncertainty estimators of each sample.

Finally, we mention [14] who developed a risk generalization bound for selective classifiers [9]. The bound presented in that paper is analogous to the coverage generalization bound we present in Theorem 4.2. The risk bound in [14] can also be used for shift-detection. To apply their risk bound to this task, however, labels, which are not available, are required. CBD detects distribution shifts without using any labels.

## 4  Proposed Method – Coverage-Based Detection

In this section we present *Coverage-Based Detection* (CBD), a novel technique for detecting a distribution shift based on selective prediction principles (definitions follow). We develop a tight generalization coverage bound that holds with high probability for ID data, sampled from the source distribution. The main idea is that violations of this coverage bound indicate a distribution shift with high probability. In Section 6.2, we offer an intuitive demonstration that aids in understanding our approach.

### 4.1  Selection with Guaranteed Coverage

We begin by introducing basic selective prediction terminology and definitions that are required to describe our method. Consider a standard multiclass classification problem, where $\mathcal{X}$ is some feature space (e.g., raw image data) and $\mathcal{Y}$ is a finite label set, $\mathcal{Y} = \{1, 2, 3, ..., C\}$, representing $C$ classes. Let $P(X, Y)$ be a probability distribution over $\mathcal{X} \times \mathcal{Y}$, and define a *classifier* as a function $f : \mathcal{X} \to \mathcal{Y}$. We refer to $P$ as the *source distribution*. A *selective classifier* [9] is a pair $(f, g)$, where $f$ is a classifier and $g : \mathcal{X} \to \{0, 1\}$ is a *selection function* [9], which serves as a binary qualifier for $f$ as follows,

$$(f, g)(x) \triangleq \begin{cases} f(x), & \text{if } g(x) = 1; \\ \text{don't know}, & \text{if } g(x) = 0. \end{cases}$$

A general approach for constructing a selection function based on a given classifier $f$ is to work in terms of a *confidence-rate function* [15], $\kappa_f : \mathcal{X} \to \mathbb{R}^+$, referred to as CF. The CF $\kappa_f$ should quantify confidence in predicting the label of $x$ based on signals extracted from $f$. The most common and well-known CF for a classification model $f$ (with softmax at its last layer) is its *softmax response* (SR) value [4, 6, 21]. A given CF $\kappa_f$ can be straightforwardly used to define a selection function: $g_\theta(x) \triangleq g_\theta(x|\kappa_f) = \mathbb{1}[\kappa_f(x) \geq \theta]$, where $\theta$ is a user-defined constant. For any selection function, we define its *coverage* w.r.t. a distribution $\mathcal{P}$ (recall, $\mathcal{P} \triangleq P_X$, see Section 2), and its *empirical coverage* w.r.t. a sample $S_k \triangleq \{x_1, x_2, \ldots x_k\}$, as $c(\theta, \mathcal{P}) \triangleq \mathbb{E}_{\mathcal{P}}[g_\theta(x)]$, and $\hat{c}(\theta, S_k) \triangleq \frac{1}{k} \sum_{i=1}^{k} g_\theta(x_i)$, respectively.

Given a bound on the expected coverage for a given selection function, we can use it to detect a distribution shift via violations of the bound. We will now formally state the problem, develop the bound, and demonstrate its use in detecting distribution shifts.

**Problem statement.** *For a classifier $f$, a detection-training sample $S_m \sim \mathcal{P}^m$, a confidence parameter $\delta > 0$, and a desired coverage $c^* > 0$, our goal is to use $S_m$ to find a $\theta$ value (which implies a selection function $g_\theta$) that guarantees the desired coverage, with probability $1 - \delta$, namely,*

$$\mathbf{Pr}_{S_m}\{c(\theta, \mathcal{P}) < c^*\} < \delta. \tag{1}$$

This means that *under coverage* should occur with probability of at most $\delta$.

A threshold $\theta$ that guarantees Equation (1) provides a probabilistic lower bound, guaranteeing that coverage $c$ of ID unseen population (sampled from $\mathcal{P}$) satisfies $c > c^*$ with probability of at least $1 - \delta$. For the remaining of this section, we introduce the *selection with guaranteed coverage* (SGC) algorithm (Algorithm 1), which outputs a bound ($b^*$) and its corresponding threshold ($\theta$).

---

**Algorithm 1:** *Selection with guaranteed coverage* (SGC)

---

1 **Input:** detection-training set: $S_m$, confidence-rate function: $\kappa_f$, confidence parameter $\delta$, target coverage: $c^*$.
2 Sort $S_m$ according to $\kappa_f(x_i)$, $x_i \in S_m$ (and now assume w.l.o.g. that indices reflect this ordering).
3 $z_{\min} = 1$, $z_{\max} = m$
4 **for** $i = 1$ **to** $k = \lceil \log_2 m \rceil$ **do**
5     $z = \lceil (z_{\min} + z_{\max})/2 \rceil$
6     $\theta_i = \kappa_f(x_z)$
7     Calculate $\hat{c}_i(\theta_i, S_m)$
8     Solve for $b_i^*(m, m \cdot \hat{c}_i(\theta_i, S_m), \frac{\delta}{k})$ {see Lemma 4.1}
9     **if** $b_i^*(m, m \cdot \hat{c}_i(\theta_i, S_m), \frac{\delta}{k}) \leq c^*$ **then**
10       $z_{\max} = z$
11     **else**
12       $z_{\min} = z$
13     **end if**
14 **end for**
15 **Output:** bound: $b_k^*(m, m \cdot \hat{c}_k(\theta_k, S_m), \frac{\delta}{k})$, threshold: $\theta_k$.

---

The SGC algorithm receives as input a classifier $f$, a CF $\kappa_f$, a confidence parameter $\delta$, a target coverage $c^*$, and a detection-training set $S_m$. The algorithm performs a binary search to find the optimal coverage lower bound with confidence $\delta$, and outputs a coverage bound $b^*$ and the corresponding threshold $\theta$, defining the selection function. A pseudo code of the SGC algorithm appears in Algorithm 1.

Our analysis of the SGC algorithm makes use of Lemma 4.1, which gives a tight numerical (generalization) bound on the expected coverage, based on a test over a sample. The proof of Lemma 4.1 is nearly identical to Langford's proof of Theorem 3.3 in [26], p. 278, where instead of the empirical error used in [26], we use the empirical coverage, which is also a Bernoulli random variable.

**Lemma 4.1.** *Let $\mathcal{P}$ be any distribution and consider a selection function $g_\theta$ with a threshold $\theta$ whose coverage is $c(\theta, \mathcal{P})$. Let $0 < \delta < 1$ be given and let $\hat{c}(\theta, S_m)$ be the empirical coverage w.r.t. the set $S_m$, sampled i.i.d. from $\mathcal{P}$. Let $b^*(m, m \cdot \hat{c}(\theta, S_m), \delta)$ be the solution of the following equation:*

$$\underset{b}{arg\,min} \left( \sum_{j=0}^{m \cdot \hat{c}(\theta, S_m)} \binom{m}{j} b^j (1-b)^{m-j} \leq 1 - \delta \right). \tag{2}$$

*Then,*

$$\mathbf{Pr}_{S_m}\{c(\theta, \mathcal{P}) < b^*(m, m \cdot \hat{c}(\theta, S_m), \delta)\} < \delta. \tag{3}$$

The following is a uniform convergence theorem for the SGC procedure stating that all the calculated bounds are valid simultaneously with a probability of at least $1 - \delta$.

**Theorem 4.2.** *(SGC – Uniform convergence) Assume $S_m$ is sampled i.i.d. from $\mathcal{P}$, and consider an application of Algorithm 1. For $k = \lceil \log_2 m \rceil$, let $b_i^*(m, m \cdot \hat{c}_i(\theta_i, S_m), \frac{\delta}{k})$ and $\theta_i$ be the values obtained in the $i^{th}$ iteration of Algorithm 1. Then,*

$$\mathbf{Pr}_{S_m}\{\exists i : c(\theta_i, \mathcal{P}) < b_i^*(m, m \cdot \hat{c}_i(\theta_i, S_m), \frac{\delta}{k})\} < \delta.$$

*Proof (sketch - see full proof in the Appendix B.1).* Define,
$\mathcal{B}_{\theta_i} \triangleq b_i^*(m, m \cdot \hat{c}_i(\theta_i, S_m), \frac{\delta}{k})$, $\mathcal{C}_{\theta_i} \triangleq c(\theta_i, \mathcal{P})$, then,

$$
\begin{aligned}
\mathbf{Pr}_{S_m}\{\exists i : \mathcal{C}_{\theta_i} < \mathcal{B}_{\theta_i}\} &= \sum_{i=1}^{k} \int_0^1 d\theta' \mathbf{Pr}_{S_m}\{\mathcal{C}_{\theta'} < \mathcal{B}_{\theta'}\} \cdot \mathbf{Pr}_{S_m}\{\theta_i = \theta'\} \\
&< \sum_{i=1}^{k} \int_0^1 d\theta' \frac{\delta}{k} \cdot \mathbf{Pr}_{S_m}\{\theta_i = \theta'\} = \sum_{i=1}^{k} \frac{\delta}{k} = \delta.
\end{aligned}
$$

$\square$

We would like to note that in addition to Algorithm 1, we also provide a complementary algorithm that returns a tight coverage **upper** (rather than lower) bound, along with the corresponding threshold. Details about this algorithm and its application for detecting distribution shifts are discussed in Appendix F.

## 4.2 Coverage-Based Detection Algorithm

Our *coverage-based detection* (CBD) algorithm applies SGC to $C_{\text{target}}$ target coverages uniformly spread between the interval $[0.1, 1]$, excluding the coverage of 1. We set $C_{\text{target}} = 10, \delta = 0.01$, and $\kappa_f(x) = 1 - \text{Entropy}(x)$ for all our experiments; our method appears to be robust to those hyper-parameters, as we demonstrate in Appendix E. Each application $j$ of SGC on the same sample $S_m \sim \mathcal{P}^m$ with a target coverage of $c_j^*$ produces a pair: $(b_j^*, \theta_j)$, which represent a bound and a threshold, respectively. We define $\delta(\hat{c}|b^*)$ to be a binary function that indicates a bound violation, where $\hat{c}$ is the empirical coverage (of a sample) and $b^*$ is a bound, $\delta(\hat{c}|b^*) = \mathbb{1}[\hat{c} \le b^*]$. Thus, given a window of $k$ samples from an alternative distribution, $W_k \sim Q^k$, we define the *sum of bound violations*, $V$, as follows,

$$
V = \frac{1}{C_{\text{target}}} \sum_{j=1}^{C_{\text{target}}} (b_j^* - \hat{c}(\theta_j, W_k)) \cdot \delta(\hat{c}(\theta_j, W_k)|b_j^*) = \frac{1}{k \cdot C_{\text{target}}} \sum_{j=1}^{C_{\text{target}}} \sum_{i=1}^{k} (b_j^* - g_{\theta_j}(x_i)) \cdot \delta(\hat{c}_j|b_j^*), \quad (4)
$$

where we obtain the last equality by using $\hat{c}_j \triangleq \hat{c}(\theta_j, W_k) = \frac{1}{k}\sum_{i=1}^{k} g_{\theta_j}(x_i)$. Considering Figure 2b in Section 6.2, the quantity $V$ is the sum of distances from the violations (red dots) to the linear diagonal representing the coverage bounds (black line).

---

**Algorithm 2: *Coverage-Based Detection***

```
1  // Fit
2  Input Training: S_m, δ, κ_f, C_target
3  Generate C_target uniformly spread coverages c*
4  for j = 1 to C_target do
5  |    b_j*, θ_j = SGC(S_m, δ, c_j*, κ_f)
6  end for
7  Output Training: {(b_j*, θ_j)}_{j=1}^{C_target}
8  // Detect
9  Input Detection: {(b_j*, θ_j)}_{j=1}^{C_target}, κ_f, α, k
10 while True do
11 |    Receive window W_k = {x_1, x_2, ..., x_k}
12 |    Calculate V {see Equation (4)}
13 |    Obtain p-value from t-test, H_0 : V = 0, H_1 : V > 0
14 |    if p_value < α then
15 |    |    Shift_detected ← True
16 |    |    Output Detection: Shift_detected, p_value
17 |    end if
18 end while
```

---

When $Q$ equals $\mathcal{P}$, which implies that there is no distribution shift, we expect that all the bounds (computed by SGC over $S_m$) will hold over $W_k$, namely $\delta(\hat{c}_j|b_j^*) = 0$ for every iteration $j$; in this case, $V = 0$. Otherwise, $V$ will indicate the violation magnitude.

Since $V$ represents the average of $k \cdot C_{\text{target}}$ values (if at least one bound violation occurs), for cases where $k \cdot C_{\text{target}} \gg 30$ (as in all our experiments), we can assume that $V$ follows a nearly normal distribution [25] and perform a t-test[1] to test the null hypothesis, $H_0 : V = 0$, against the alternative hypothesis, $H_1 : V > 0$. The null hypothesis is rejected if the p-value is less than the specified significance level $\alpha$. A pseudocode of our *coverage-based detection algorithm* appears in Algorithm 2.

To fit our detector, we apply SGC (Algorithm 1) on the detection-training set, $S_m$, for $C_{\text{target}}$ times, in order to construct the pairs $\{(b_j^*, \theta_j)\}_{j=1}^{C_{\text{target}}}$. Our detection model utilizes these pairs to monitor a given model, receiving at each time instant a test sample window of size $k$ (user defined), $W_k = \{x_1, x_2, \ldots, x_k\}$, which is inspected to see if its content is distributionally shifted from the underlying distribution reflected by the detection-training set $S_m$.

---

[1]In our experiments, we use SciPy's `stats.ttest_1samp` implementation [44] for the t-test.

Our approach encodes all necessary information for detecting distribution shifts using only $C_{\text{target}}$ scalars (in our experiments, we set $C_{\text{target}} = 10$), which is independent of the size of $S_m$. In contrast, the baselines process the detection-training set, $S_m$, which is typically very large, for every detection they make. This makes our method significantly more efficient than the baselines, see Figure 1 and Table 1 in Section 5.

## 5  Complexity Analysis

This section provides a complexity analysis of our method as well as the baselines mentioned in Section 3. Table 1 summarizes the complexities of each approach. Figure 1 shows the population-based approaches run-time (in seconds) as a function of the detection-training set size, denoted as $m$.

All population-based baselines are lazy learners (analogous to nearest neighbors) in the sense that they require the entire source (detection-training) set for each detection decision they make. Using only a subset will result in sub-optimal performance, since it might not capture all the necessary information within the source distribution, $\mathcal{P}$.

| Detection Method | Space | Time |
|---|---|---|
| MMD | $\mathcal{O}(m^2 + k^2 + mk)$ | $\mathcal{O}(d(m^2 + k^2 + mk))$ |
| KS | $\mathcal{O}(d(m + k))$ | $\mathcal{O}(d(m \log m + k \log k))$ |
| Single-instance | $\mathcal{O}(k)$ | $\mathcal{O}(k)$ |
| **CBD** (Ours) | $\mathcal{O}(k)$ | $\mathcal{O}(k)$ |

Table 1: Complexity comparison. **bold** entries indicate the best detection complexity. $m$, $k$ refers to the detection-training size (or source size), and window size, respectively. $d$ refers to the number of dimensions after dimensionality reduction.

In particular, MMD is a permutation test [18] that also employs a kernel. The complexity of kernel methods is dominated by the number of instances and, therefore, the time and space complexities of MMD are $\mathcal{O}(d(m^2 + k^2 + mk))$ and $\mathcal{O}(m^2 + k^2 + mk)$, respectively [2], where in the case of DNNs, $d$ is the dimension of the embedding or softmax layer used for computing the kernel, and $k$ is the window size. The KS test [33] is a univariate test, which is applied on each dimension separately and then aggregates the results via a Bonferroni correction. Its time and space complexities are $\mathcal{O}(d(m \log m + k \log k))$ and $\mathcal{O}(d(m + k))$, respectively.

The single-instance baselines simply conduct a t-test on the heuristic uncertainty estimators (SR or Entropy-based) between the detection-training set and the window data. By initially determining the mean and standard deviation of the detection-training set, we can efficiently perform the t-test with a time and space complexity of $\mathcal{O}(k)$.

The fitting procedure of our coverage-based detection algorithm incurs time and space complexities of $\mathcal{O}(m \log m)$ and $\mathcal{O}(m)$, respectively. Each subsequent detection round incurs time and space complexities of $\mathcal{O}(k)$, which is independent of the size of the detection-training (or source) set. Our method's superior efficiency is demonstrated both theoretically and empirically, as shown in Table 1 and Figure 1, respectively. Figure 1 shows the run-time (in seconds) for each population-based detection method[3] as a function of the detection-training set size, using a ResNet50 and a fixed window size of $k = 10$ samples. Our detection time remains constant regardless of the size of the detection-training set, as opposed to the baseline methods, which exhibit a run-time that grows as a function of the detection-training set size. In particular, our method achieves a five orders of magnitude improvement in run-time over the best performing baseline, KS-Softmax, when the detection-training set size is 1,000,000[4]. Specifically, our detection time is $5.19 \cdot 10^{-4}$ seconds, while KS-Softmax's detection time is 97.1 seconds, rendering it inapplicable for real-world applications.

## 6  Experiments - Detecting Distribution Shifts

In this section we showcase the effectiveness of our method, along with the considered baselines, in detecting distribution shifts. All experiments were conducted using PyTorch, and the corresponding code is publicly available for replicating our results[5].

---

[2]With preprocessing in the fitting stage, this reduces to $\mathcal{O}(d(k^2 + mk))$ per detection decision.

[3]MMD detection-training set is capped at 1,000 due to kernel method's unfavorable size dependence.

[4]This specific experiment is simulated synthetically (since the ImageNet validation set size is 50,000).

[5] `https://github.com/BarSGuy/Window-Based-Distribution-Shift-Detection`.

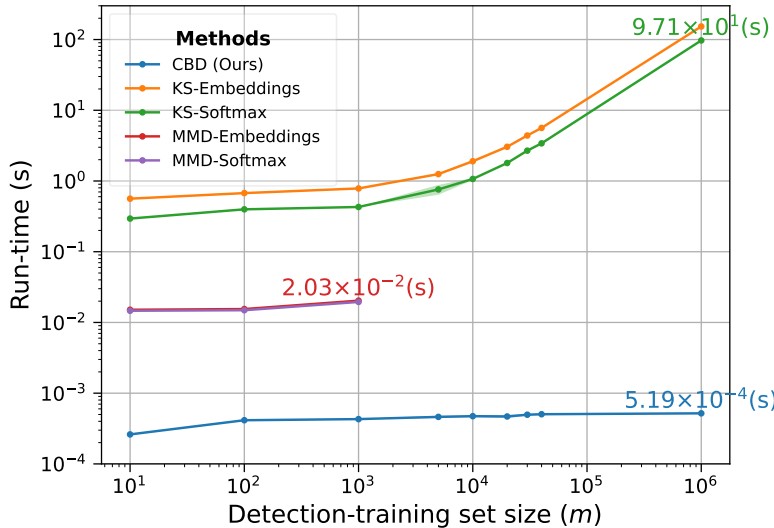

Figure 1: Our method outperforms the baselines in terms of scalability, with a significant five orders of magnitude improvement in run-time compared to the best baseline, KS-Softmax, when using a detection-training set of $m = 1,000,000$ samples. One-$\sigma$ error-bars are shadowed.

## 6.1 Setup

Our experiments are conducted on the ImageNet dataset [7], using its validation dataset as proxies for the source distribution, $\mathcal{P}$. We utilized three well-known architectures, ResNet50 [19], MobileNetV3-S [24], and ViT-T [8], all of which are publicly available in timm's repository [45], as our pre-trained models. To train our detectors, we randomly split the ImageNet validation data (50,000) into two sets, a detection-training or source set, which is used to fit the detectors[3] (49,000) and a validation set (1,000) for applying the shift. To ensure the reliability of our results, we repeated the shift detection performance evaluation on 15 random splits, applying the same type of shift to different subsets of the data. Inspired by [37], we evaluated the models using various window sizes, $|W_k| \in \{10, 20, 50, 100, 200, 500, 1000\}$.

### 6.1.1 Distribution Shift Datasets

At test time, the 1,000 validation images obtained via our split can be viewed as the in-distribution (positive) examples. For out-of-distribution (negative) examples, we follow the common setting for detecting OOD samples or distribution shifts [37, 21, 28], and test on several different natural image datasets and synthetic noise datasets. More specifically, we investigate the following types of shifts:

(1) **Adversarial via FGSM**: We transform samples into adversarial ones using the *Fast Gradient Sign Method* (FGSM) [17], with $\epsilon \in \{7 \cdot 10^{-5}, 1 \cdot 10^{-4}, 3 \cdot 10^{-4}, 5 \cdot 10^{-4}\}$. (2) **Adversarial via PGD**: We convert samples into adversarial examples using *Projected Gradient Descent* (PGD) [31], with $\epsilon = 1 \cdot 10^{-4}$; we use 10 steps, $\alpha = 1 \cdot 10^{-4}$, and random initialization. (3) **Gaussian noise**: We corrupt test samples with Gaussian noise using standard deviations of $\sigma \in \{0.1, 0.3, 0.5, 1\}$. (4) **Rotation**: We apply image rotations, $\theta \in \{5°, 10°, 20°, 25°\}$. (5) **Zoom**: We corrupt test samples by applying zoom-out percentages of $\{90\%, 70\%, 50\%\}$. (6) **ImageNet-O**: We use the ImageNet-O dataset [23], consisting of natural out-of-distribution samples. (7) **ImageNet-A**: We use the ImageNet-A dataset [23], consisting of natural adversarial samples. (8) **No-shift**: We include the no-shift case to check for false positives. To determine the severity of the distribution shift, we refer the reader to Table 3 in Appendix C.

## 6.2 Maximizing Detection Power Through Lower Coverages

In this section, we provide an intuitive understanding of our proposed method, as well as a demonstration of the importance of considering lower coverages, when detecting population-based distribution shifts. For all experiments in this section, we employed a ResNet50 [19] and used a detection-training set which was randomly sampled from the ImageNet validation set.

We validate the effectiveness of our CBD method (Algorithm 2) by demonstrating its performance on two distinct scenarios, the no-shift case and the shift case. In the no-shift scenario, as depicted in Figure 2a, we randomly

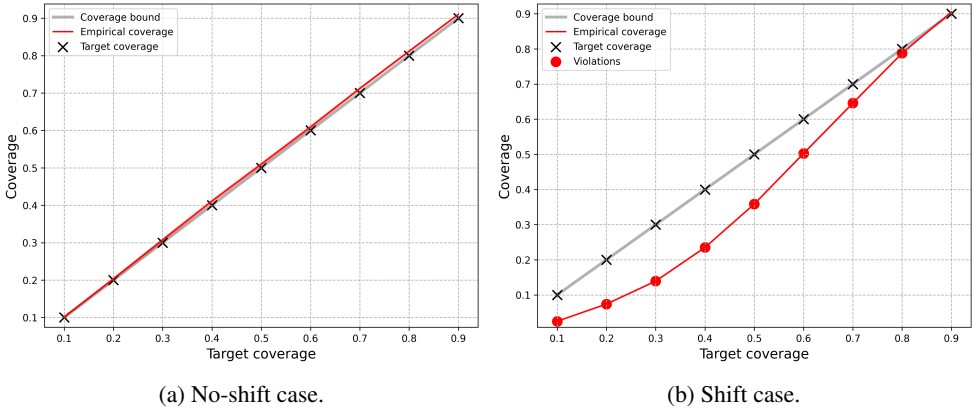

(a) No-shift case.          (b) Shift case.

Figure 2: Visualization of our detection algorithm, Algorithm 2.

selected a 1,000 sample window from the ImageNet dataset that did not overlap with the detection-training set. We then tested whether this window was distributionally shifted using our method. The shift scenario, as depicted in Figure 2b, involved simulating a distributional shift by randomly selecting 1,000 images from the ImageNet-O dataset. Each figure showcases the target coverages of each application, the coverage bounds provided by `SGC`, the empirical coverage of each sample (for each threshold), and the bound violations (indicated by the red circles), when they occur.

Both figures show that the target coverages and the bounds returned by `SGC` are essentially identical. Consider the empirical coverage of each threshold returned by `SGC` for each sample. In the no-shift case, illustrated in Figure 2a, it is apparent that all bounds are tightly held; demonstrated by the fact that each empirical coverage is slightly above its corresponding bound. In contrast, the shift case, depicted in Figure 2b, exhibits bound violations, indicating a noticeable distribution shift. Specifically, the bound holds at a coverage of 0.9, but violations occur at lower coverages, with the maximum violation magnitude occurring around a coverage of 0.4. This underscores the significance of using coverages lower than 1, and suggests that the detection power might lay within lower coverages.

## 6.3 Evaluation Metrics

In order to benchmark our results against the baselines, we adopt the following metrics to measure the effectiveness in distinguishing between in- and out-of-distribution windows: (1) **Area Under the Receiver Operating Characteristic** ↑ (AUROC) is a threshold-independent metric [5]. The ROC curve illustrates the relationship between the *true positive rate* (TPR) and the *false positive rate* (FPR), and the AUROC score represents the probability that a positive example will receive a higher detection score than a negative example [10]. A perfect detector would achieve an AUROC of 100%. (2) **Area Under the Precision-Recall** ↑ (AUPR) is another threshold-independent metric [32, 40]. The PR curve is a graph that depicts the relationship between *precision* (TP / (TP + FP)) and *recall* (TP / (TP + FN)). AUPR-In and AUPR-Out in Table 2 represent the area under the precision-recall curve where in-distribution and out-of-distribution images are considered as positives, respectively. (3) **False positive rate (FPR) at 95% true positive rate (TPR)** ↓, denoted as FPR@95TPR, is a performance metric that measures the FPR at a specific TPR threshold of 95%. This metric is calculated by finding the FPR when the TPR is 95%. (4) **Detection Error** ↓ is the probability of misclassification when TPR is fixed at 95%. It's a weighted average of FPR and complement of TPR, given by $P_e = 0.5(1-TPR)+0.5FPR$. Assuming equal probability of positive and negative examples in the test set. We note that ↑ indicates larger value is better, and ↓ indicates lower values is better.

## 6.4 Experimental Results

In this section we demonstrate the effectiveness of our CBD method, as well as the considered baselines. To ensure a fair comparison, as mentioned in Section 6.1 ,we utilize the ResNet50, MobileNetV3-S, and ViT-T architectures, which are pretrained on ImageNet and their open-sourced code appear at timm [45]. We provide code for reproducing our experiments[5].

Table 2 presents a comprehensive summary of our main results, which clearly demonstrate that CBD outperforms all baseline methods in the majority of cases (combinations of model architecture, window size, and metrics considered), thus providing a clear indication of the effectiveness of our approach. Our method demonstrates the highest effectiveness in the low to mid window size regime. Specifically, for window sizes of 50 samples,

our method outperforms all baselines across all architectures, with statistical significance for all considered metrics, by a large margin. This observation is true for window sizes of 100 samples as well, except for the MobileNet architecture, where KS-Softmax shows slightly better performance in the metrics of Detection Error and FPR@95TPR. The second-best performing method overall appears to be KS-Softmax (consistent with the findings of [37]), which seems to be particularly effective in large window sizes, such as 500 or more. CBD stands out for its consistent performance across different architectures. In particular, our method achieves an AUROC score of over 90% across all architectures tested when using a window size of 50+, while the second best performing method (KS-Softmax), fails to achieve such a score even with a window size of 100 samples, when using the ResNet50 architecture.

It is evident from Table 2, that as far as population-based detection tasks are concerned, neither of the single-instance baselines (SR and Entropy) offers an effective solution. These baselines show inconsistent performance across the various architectures considered. For instance, when ResNet50 is used over a window size of 1,000 samples, the baselines scarcely achieve a score of 90% for the threshold-independent metrics (AUROC, AUPR-In, and AUPR-Out), while other methods achieve near-perfect scores. Moreover, even in the low sample regime (e.g., window size of 10), these baselines fail to demonstrate any superiority over population-based methods.

Finally, based on our experiments, ResNet50 appears to be the best model for achieving low Detection Error and FPR@95TPR; and a large window size is essential for achieving these results. In particular, our method and KS-Softmax both produced impressive outcomes, with Detection Error and FPR@95TPR scores under 0.5 and 1, respectively, when a window size of 1,000 samples was employed. Moreover, we observe that when small window sizes are desired, ViT-T is the best architecture choice. In particular, our method applied with ViT-T, achieves a phenomenal 86%+ score in all threshold independent metrics (i.e., AUROC, AUPR-In, AUPR-Out), over a window size of 10 samples, strongly out-preforming the contenders (around 20% margin). Appendix D presents the detection performance on a representative selection of distribution shifts, analyzed individually.

| Architecture | Method | | Window size AUROC ↑ / AUPR-In ↑ / AUPR-Out ↑ / DetectionError ↓ / FPR@95TPR ↓ | | | | | | |
|---|---|---|---|---|---|---|---|---|---|
| | | | 10 | 20 | 50 | 100 | 200 | 500 | 1000 |
| ResNet50 | KS | Softmax | 61/67/62/34/67 | 73/74/74/31/64 | 87/90/85/13/27 | 89/89/89/15/29 | 94/95/92/7/14 | **99/99/99/2/4***  | **100/100/100/0.4/0.9** |
| | | Embeddings | 72/**74**/73/**28***/**56*** | 68/73/74/24/48 | 81/84/79/18/37 | 75/76/79/22/44 | 76/79/79/20/40 | 84/87/84/13/26 | 86/88/84/13/26 |
| | MMD | Softmax | 54/61/56/36/72 | 62/65/62/37/72 | 73/76/72/29/56 | 73/73/78/33/59 | 79/79/79/35/54 | 83/85/83/15/30 | 85/85/85/22/37 |
| | | Embeddings | 75/72/77/38/70 | 79/78/79/29/57 | 87/87/86/18/37 | 83/86/81/15/30 | 83/85/82/14/29 | 83/85/83/17/32 | 83/86/82/13/26 |
| | Single-instance | SR | 56/65/55/34/68 | 72/73/72/32/63 | 71/75/72/28/56 | 77/78/79/25/50 | 84/85/83/19/40 | 87/88/87/14/28 | 88/88/89/15/30 |
| | | Entropy | 64/69/63/32/64 | 73/73/73/32/63 | 74/78/73/26/52 | 80/80/81/23/47 | 84/85/84/17/35 | 87/87/87/15/31 | 90/90/91/13/26 |
| | CBD (Ours) | | **78**/70/**82***/42/64 | **88***/**91***/**87***/**15***/**30*** | **95***/**95***/**93***/**9***/**17*** | **93***/**93***/**92***/**10***/**20*** | **97***/**97***/**97***/**5**/**10** | 98/98/98/4/7 | **100/100/100/0.4/0.9** |
| MobileNetV3-S | KS | Softmax | 71/72/75/**32***/**63*** | **84***/84/83/21/43 | 89/91/88/13/27 | 92/93/91/10/28 | **95***/**97***/94*/5/11 | 96/96/**97**/6/11 | **100/100/100/1/2** |
| | | Embeddings | 63/67/63/37/75 | 65/66/67/37/75 | 77/78/76/27/54 | 72/73/76/27/53 | 84/83/86/22/43 | 86/87/86/15/30 | 79/81/81/18/36 |
| | MMD | Softmax | 75/**73**/75/38/72 | 78/78/78/30/59 | 86/89/82/17/32 | 86/89/84/14/26 | 87/88/86/14/28 | 89/90/88/12/24 | 90/91/88/11/22 |
| | | Embeddings | 67/67/68/39/75 | 66/67/68/37/74 | 72/77/71/23/47 | 75/75/79/28/53 | 89/87/87/20/39 | 81/82/81/21/40 | 82/86/80/15/30 |
| | Single-instance | SR | 58/62/60/37/74 | 65/70/66/30/60 | 86/87/86/19/39 | 86/88/84/15/30 | 93/93/93/11/22 | 96/97/95/5/10 | 98/98/97/3/7 |
| | | Entropy | 52/61/57/36/72 | 64/70/65/29/57 | 86/87/85/15/29 | 87/88/85/15/29 | 93/93/94/11/22 | 96/96/96/6/12 | 98/98/98/3/7 |
| | CBD (Ours) | | **80***/**73**/**82***/40/80 | 80/**85**/76/**20**/**40** | **94***/**95***/**93***/**8***/**15*** | **94**/**94**/**94***/11/21 | 95/96/95/6/13 | **97**/**98**/96/**4**/**8** | 99/99/99/**1**/**2** |
| ViT-T | KS | Softmax | 62/68/66/30/59 | 85/86/83/19/41 | 82/82/83/21/42 | 90/91/91/11/22 | 88/89/90/11/22 | 95/96/95/5/11 | 98/98/98/3/6 |
| | | Embeddings | 68/66/71/38/76 | 76/83/73/19/**38** | 81/82/81/20/39 | 81/82/86/17/34 | 81/84/80/17/34 | 76/75/82/22/44 | 84/83/86/19/38 |
| | MMD | Softmax | 58/59/64/44/82 | 69/70/73/38/69 | 77/80/77/22/44 | 75/80/76/20/40 | 80/86/78/15/29 | 89/91/90/12/23 | 93/94/92/7/15 |
| | | Embeddings | 61/59/66/46/85 | 74/77/74/26/53 | 77/78/77/22/44 | 80/82/83/21/39 | 82/81/82/20/40 | 78/76/81/22/44 | 77/78/79/24/45 |
| | Single-instance | SR | 67/70/70/31/61 | 78/76/77/29/58 | 76/79/76/23/45 | 89/90/86/14/29 | 91/93/91/9/20 | **99**/99/98/2/5 | **99**/**99**/98/2/5 |
| | | Entropy | 69/74/69/**27**/**53** | 79/78/78/27/55 | 75/78/74/22/44 | 89/89/90/15/31 | 86/87/87/14/28 | 93/94/93/7/14 | 97/97/97/4/9 |
| | CBD (Ours) | | **89***/**86***/**90***/28/56 | **91***/**87**/**92***/**24**/47 | **94***/**93***/**95***/**13***/**25*** | **95***/**96***/**96***/**8***/**15*** | **97***/**96***/**97***/**8**/16 | **98***/**98**/**98**/4/9 | **99/99/99**/3/6 |

Table 2: Comparison of different evaluation metrics over **ResNet50**, **MobileNetV3-S**, **ViT-T** with the discussed baselines methods. The best performing method is highlighted in **bold**; we add the superscript * to the bolded result when it is statistically significant.

# 7  Concluding Remarks

We presented a novel and powerful method for the detection of distribution shifts within a given window of samples. This coverage-based detection algorithm is theoretically motivated and can be applied to any pretrained model. Due to its low computational complexity, our method, unlike typical baselines, which are order-of-magnitude slower, is practicable. Our comprehensive empirical studies demonstrate that the proposed method works very well, and overall significantly outperforms the baselines on the ImageNet dataset, across a number of neural architectures and a variety of distribution shifts, including adversarial examples. In addition, our coverage bound is of independent interest and allows for the creation of selective classifiers with guaranteed coverage.

Several directions for future research are left open. Although we only considered classification, our method can be extended to regression using an appropriate confidence-rate function such as the MC-dropout [12]. Extensions to other tasks, such as object detection and segmentation, would be very interesting. In our method, the information from the multiple coverage bounds was aggregated by averaging, but it is plausible that other statistics or weighted averages could provide more effective detections. Finally, an interesting open question is whether one can benefit from using single-instance or outlier/adversarial detection techniques combined with population-based detection techniques (as discussed here).

## Acknowledgments

This research was partially supported by the Israel Science Foundation, grant No. 710/18.

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

# A  Shift-Detection General Framework

The general framework for shift-detection can be found in the following figure, Figure 3.

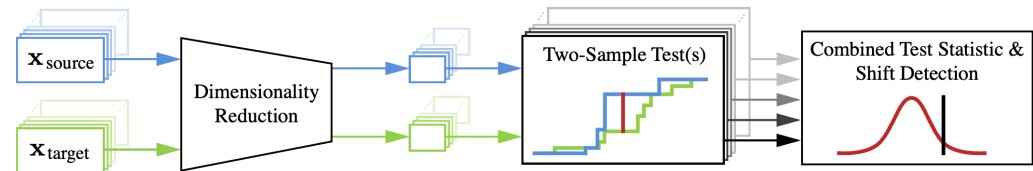

Figure 3: The procedure of detecting a dataset shift using dimensionality reduction and then a two-sample statistical test. The dimensionality reduction is applied to both the detection-training (source) and test (target) data, prior to being analyzed using statistical hypothesis testing. This figure is taken from [37].

# B  Proofs

## B.1  Proof for Theorem 4.2

*Proof.* Define

$$\mathcal{B}_{\theta_i} \triangleq b_i^*(m, m \cdot \hat{c}_i(\theta_i, S_m), \frac{\delta}{k}),$$

$$\mathcal{C}_{\theta_i} \triangleq c(\theta_i, \mathcal{P}).$$

Consider the i-th iteration of SGR over a detection-training set $S_m$, and recall that, $\theta_i = \kappa_f(x_z)$, $x_z \in S_m$ (see Algorithm 1). Therefore, $\theta_i$ is a random variable (between zero and one), since it is a function of a random variable ($x \in S_m$). Let $\mathbf{Pr}_{S_m}\{\theta_i = \theta'\}$ be the probability that $\theta_i = \theta'$.

Therefore,

$$\mathbf{Pr}_{S_m}\{\mathcal{C}_{\theta_i} < \mathcal{B}_{\theta_i}\}$$
$$= \int_0^1 d\theta' \mathbf{Pr}_{S_m}\{\mathcal{C}_{\theta_i} < \mathcal{B}_{\theta_i} | \theta_i = \theta'\} \cdot \mathbf{Pr}_{S_m}\{\theta_i = \theta'\}$$
$$= \int_0^1 d\theta' \mathbf{Pr}_{S_m}\{\mathcal{C}_{\theta'} < \mathcal{B}_{\theta'}\} \cdot \mathbf{Pr}_{S_m}\{\theta_i = \theta'\}.$$

Given that at the i-th iteration $\theta_i = \theta'$, and considering that $\mathcal{B}_{\theta'}$ is derived using Lemma 4.1 (refer to Algorithm 1), we obtain,

$$\mathbf{Pr}_{S_m}\{\mathcal{C}_{\theta'} < \mathcal{B}_{\theta'}\} < \frac{\delta}{k}.$$

Thus,

$$\mathbf{Pr}_{S_m}\{\mathcal{C}_{\theta_i} < \mathcal{B}_{\theta_i}\}$$
$$= \int_0^1 d\theta' \mathbf{Pr}_{S_m}\{\mathcal{C}_{\theta'} < \mathcal{B}_{\theta'}\} \cdot \mathbf{Pr}_{S_m}\{\theta_i = \theta'\}$$
$$< \int_0^1 d\theta' \frac{\delta}{k} \cdot \mathbf{Pr}_{S_m}\{\theta_i = \theta'\}$$
$$= \frac{\delta}{k} \cdot \left( \int_0^1 d\theta' \mathbf{Pr}_{S_m}\{\theta_i = \theta'\} \right)$$
$$= \frac{\delta}{k}. \tag{5}$$

The following application of the union bound completes the proof,

$$\mathbf{Pr}_{S_m}\{\exists i : \mathcal{C}_{\theta_i} < \mathcal{B}_{\theta_i}\} \leq \sum_{i=1}^k \mathbf{Pr}_{S_m}\{\mathcal{C}_{\theta_i} < \mathcal{B}_{\theta_i}\} < \sum_{i=1}^k \frac{\delta}{k} = \delta.$$

□

# C   Exploring Model Sensitivity: Evaluating Accuracy on Shifted Datasets

In this section, we present Table 3, which displays the accuracy (when applicable) as well as the degradation from the original accuracy over the ImageNet dataset, of the considered models on each of the simulated shifts mentioned in Section 6.1.1.

| Shift Dataset | ResNet50 | | MovileNetV3 | | ViT-T | |
|---|---|---|---|---|---|---|
| | Acc. | ImageNet Degradation | Acc. | ImageNet Degradation | Acc. | ImageNet Degradation |
| FGSM $\epsilon = 7 \cdot 10^{-5}$ | 76.68% | -3.7% | 62.09% | -3.15% | 72.51% | -2.95% |
| FGSM $\epsilon = 1 \cdot 10^{-4}$ | 75.19% | -5.19% | 60.72% | -4.52% | 71.49% | -3.97% |
| FGSM $\epsilon = 3 \cdot 10^{-4}$ | 66.15% | -14.23% | 52.09% | -13.15% | 65.06% | -10.4% |
| FGSM $\epsilon = 5 \cdot 10^{-4}$ | 59.23% | -21.15% | 44.45% | -20.79% | 58.9% | -16.56% |
| PGD $\epsilon = 1 \cdot 10^{-4}$ | 74.64% | -5.74% | 60.63% | -4.61% | 71.35% | -4.11% |
| GAUSSIAN $\sigma = 0.1$ | 79.02% | -1.36% | 62.82% | -2.42% | 71.79% | -3.67% |
| GAUSSIAN $\sigma = 0.3$ | 74.63% | -5.75% | 55.06% | -10.18% | 50.86% | -24.6% |
| GAUSSIAN $\sigma = 0.5$ | 68.56% | -11.82% | 42.55% | -22.69% | 22.25% | -53.21% |
| GAUSSIAN $\sigma = 1$ | 46.1% | -34.28% | 13.82% | -51.42% | 0.56% | -74.9% |
| ZOOM 50% | 65.55% | -14.83% | 36.96% | -28.28% | 46.04% | -29.42% |
| ZOOM 70% | 74.31% | -6.07% | 53.53% | -11.71% | 62.69% | -12.77% |
| ZOOM 90% | 78.6% | -1.78% | 61.28% | -3.96% | 72.08% | -3.38% |
| ROTATION $\theta = 5°$ | 76.7% | -3.68% | 62.42% | -2.82% | 71.27% | -4.19% |
| ROTATION $\theta = 10°$ | 72.4% | -7.98% | 58.22% | -7.02% | 67.29% | -8.17% |
| ROTATION $\theta = 20°$ | 68.29% | -12.09% | 49.96% | -15.28% | 62.38% | -13.08% |
| ROTATION $\theta = 25°$ | 70.08% | -10.3% | 50.95% | -14.29% | 60.97% | -14.49% |

Table 3: Shifted dataset accuracy and comparison with ImageNet. We display the accuracy for each shifted dataset and model combination, along with the accuracy degradation when compared to the original ImageNet dataset.

## D   Extended Empirical Results

In this section, we present a detailed analysis of our empirical findings on the ResNet50 architecture. We report the results for each window size, $|W_k| \in \{10, 20, 50, 100, 200, 500, 1000\}$, and for several shift cases discussed in Section 6.1.1. In particular, we show the detection performance of all the discussed methods, for the following shifts: FGSM (Table 4), ImageNet-O (Table 5), ImageNet-A (Table 6), and the Zoom out shift, 90% (Table 7).

| Method | | Window size | | | | | | |
|---|---|---|---|---|---|---|---|---|
| | | AUROC ↑ / AUPR-In ↑ / AUPR-Out ↑ / DetectionError ↓ / FPR@95TPR ↓ | | | | | | |
| | | 10 | 20 | 50 | 100 | 200 | 500 | 1000 |
| KS | Softmax | 32/45/40/47/92 | 47/55/46/44/91 | 64/72/59/34/69 | 72/72/75/38/77 | 80/87/69/18/36 | **100/100/100**/2/4 | **100/100/100**/0/0 |
| | Embeddings | 54/58/55/**43**/86 | 32/39/48/49/100 | 54/64/49/39/80 | 41/44/48/50/99 | 37/48/45/47/92 | 60/70/59/30/60 | 71/77/61/33/68 |
| MMD | Softmax | 36/48/42/45/90 | 44/56/44/42/82 | 51/53/51/48/93 | 41/44/54/49/97 | 50/52/50/48/94 | 48/52/51/45/93 | 55/55/55/47/94 |
| | Embeddings | 61/56/60/48/95 | 57/57/59/45/93 | 72/73/67/36/73 | 63/70/56/38/71 | 63/69/55/37/75 | 67/70/61/39/74 | 70/79/59/28/54 |
| Single-instance | SR | 34/45/40/47/93 | 69/68/72/42/82 | 43/52/50/45/90 | 54/54/61/47/93 | 62/64/58/42/86 | 66/73/59/35/72 | 72/69/73/43/86 |
| | Entropy | 42/49/44/47/94 | 65/60/65/47/92 | 49/55/49/45/89 | 59/53/63/49/98 | 60/66/58/39/77 | 59/59/56/45/90 | 64/61/63/46/90 |
| CBD (Ours) | | **71/64/75**/45/92 | **77/82/75/25/51\*** | **88/90/85/20/39** | **84/86/84/25/49** | **99\*/99\*/99\*/3\*/5\*** | 98/98/98/5/10 | **100/100/100**/2/2 |

Table 4: Comparison of different evaluation metrics over **ResNet50** with the discussed baselines methods, over the FGSM shift with $\epsilon = 0.0001$. The best performing method is highlighted in **bold**; we add the superscript * to the bolded result when it is statistically significant.

| Method | | Window size | | | | | | |
|---|---|---|---|---|---|---|---|---|
| | | AUROC ↑ / AUPR-In ↑ / AUPR-Out ↑ / DetectionError ↓ / FPR@95TPR ↓ | | | | | | |
| | | 10 | 20 | 50 | 100 | 200 | 500 | 1000 |
| KS | Softmax | 62/59/60/46/94 | 70/61/75/48/94 | 98/98/98/6/11 | 99/99/99/5/10 | **100/100/100/0/0** | **100/100/100/0/0** | **100/100/100/0/0** |
| | Embeddings | 85/89/76/18/35 | **97/98/97/6\*/12\*** | 99/99/99/5/9 | **100/100/100/0/0** | **100/100/100/0/0** | **100/100/100/0/0** | **100/100/100/0/0** |
| MMD | Softmax | 43/53/47/44/87 | 74/75/73/39/72 | 94/92/96/33/51 | 97/97/98/31/27 | 97/97/98/32/26 | **100/100/100/0/0** | **100/100/100/0/0** |
| | Embeddings | **96/97/97\*/10/20** | 95/94/97/33/39 | **100/100/100/0/0** | **100/100/100/0/0** | **100/100/100/0/0** | 94/95/97/31/46 | **100/100/100/0/0** |
| Single-instance | SR | 62/63/60/43/87 | 31/41/38/48/98 | 47/56/44/41/86 | 56/57/63/45/85 | 51/51/53/48/97 | 37/41/42/50/100 | 42/44/47/49/100 |
| | Entropy | 64/66/68/40/81 | 39/42/53/50/99 | 41/52/45/44/90 | 58/66/52/37/75 | 54/51/55/49/99 | 52/55/53/48/90 | 84/85/84/26/52 |
| CBD (Ours) | | 61/61/61/47/92 | 84/89/77/18/37 | 99/99/99/5/8 | **100/100/100/0/0** | **100/100/100/0/0** | **100/100/100/0/0** | **100/100/100/0/0** |

Table 5: Comparison of different evaluation metrics over **ResNet50** with the discussed baselines methods, over the ImageNet-O shift. The best performing method is highlighted in **bold**; we add the superscript * to the bolded result when it is statistically significant.

| Method | | Window size | | | | | | |
|---|---|---|---|---|---|---|---|---|
| | | AUROC ↑ / AUPR-In ↑ / AUPR-Out ↑ / DetectionError ↓ / FPR@95TPR ↓ | | | | | | |
| | | 10 | 20 | 50 | 100 | 200 | 500 | 1000 |
| KS | Softmax | **100/100/100/0/0** | **100/100/100/0/0** | **100/100/100/0/0** | **100/100/100/0/0** | **100/100/100/0/0** | **100/100/100/0/0** | **100/100/100/0/0** |
| | Embeddings | 98/98/98/7/14 | **100/100/100/0/0** | **100/100/100/0/0** | **100/100/100/0/0** | **100/100/100/0/0** | **100/100/100/0/0** | **100/100/100/0/0** |
| MMD | Softmax | **100/100/100/0/0** | **100/100/100/0/0** | **100/100/100/0/0** | **100/100/100/0/0** | **100/100/100/0/0** | **100/100/100/0/0** | **100/100/100/0/0** |
| | Embeddings | **100/100/100/0/0** | **100/100/100/0/0** | **100/100/100/0/0** | **100/100/100/0/0** | **100/100/100/0/0** | **100/100/100/0/0** | **100/100/100/0/0** |
| Single-instance | SR | **100/100/100/0/0** | **100/100/100/0/0** | **100/100/100/0/0** | **100/100/100/0/0** | **100/100/100/0/0** | **100/100/100/0/0** | **100/100/100/0/0** |
| | Entropy | **100/100/100/0/0** | **100/100/100/0/0** | **100/100/100/0/0** | **100/100/100/0/0** | **100/100/100/0/0** | **100/100/100/0/0** | **100/100/100/0/0** |
| CBD (Ours) | | **100/100/100/0/0** | **100/100/100/0/0** | **100/100/100/0/0** | **100/100/100/0/0** | **100/100/100/0/0** | **100/100/100/0/0** | **100/100/100/0/0** |

Table 6: Comparison of different evaluation metrics over **ResNet50** with the discussed baselines methods, over the ImageNet-A shift. The best performing method is highlighted in **bold**; we add the superscript * to the bolded result when it is statistically significant.

| | **Method** | Window size | | | | | | |
|---|---|---|---|---|---|---|---|---|
| | | AUROC ↑ / AUPR-In ↑ / AUPR-Out ↑ / DetectionError ↓ / FPR@95TPR ↓ | | | | | | |
| | | 10 | 20 | 50 | 100 | 200 | 500 | 1000 |
| KS | Softmax | 42/49/48/47/93 | 53/52/54/47/97 | 61/69/57/37/74 | 71/72/69/39/77 | **91**/**92**/**91**\*/15/30 | **100**\*/**100**\*/**100**\*/**2**\*/**3**\* | **100**/**100**/**100**/**0**/**0** |
| | Embeddings | 46/51/52/46/92 | 25/45/38/44/87 | 56/58/54/44/86 | 46/46/53/49/98 | 35/44/42/49/97 | 27/38/38/50/99 | 41/46/46/47/97 |
| MMD | Softmax | 48/51/50/**46**/93 | 46/44/52/50/100 | 51/57/52/45/88 | 50/56/52/46/88 | 52/57/49/46/88 | 51/57/47/44/90 | 53/53/54/49/96 |
| | Embeddings | 57/56/61/48/**91** | 54/63/49/41/83 | 68/67/73/40/82 | 37/53/41/43/83 | 28/41/38/49/97 | 31/38/41/50/100 | 18/35/34/49/100 |
| Single-instance | SR | 28/38/38/50/100 | 45/45/48/51/99 | 44/53/53/43/87 | 52/59/53/42/84 | 65/68/59/41/85 | 74/73/78/38/78 | 84/84/83/31/62 |
| | Entropy | 29/38/39/50/100 | 50/47/56/51/100 | 51/53/58/46/91 | 61/63/66/41/83 | 71/74/61/36/72 | 76/73/81/39/77 | 87/86/87/29/58 |
| CBD (Ours) | | **70**/**61**/**77**/47/95 | **69**/**73**/**68**/35/70 | **81**/**86**/**75**/**22**/**43** | **72**/**75**/**71**/34/67 | 76/82/71/23/46 | 94/94/94/15/31 | **100**/**100**/**100**/**0**/**0** |

Table 7: Comparison of different evaluation metrics over **ResNet50** with the discussed baselines methods, over the Zoom out (90%) shift. The best performing method is highlighted in **bold**; we add the superscript \* to the bolded result when it is statistically significant.

# E Ablation Study

In this section, we conduct multiple experiments to analyze the various components of our framework; all those experiments are conducted using a ResNet50. We explore several hyper-parameter choices, including $C_{\text{target}}$, $\delta$, and $\kappa_f$. More specifically, we consider $C_{\text{target}} \in \{1, 10, 100\}$, and $\delta \in \{0.1, 0.01, 0.001, 0.0001\}$, and two different CFs $\kappa_f$, namely SR and Entropy-based.

To evaluate the performance of our detectors under varying hyper-parameters, we have selected a single metric that we believe to be the most important, namely, AUROC [13]. Additionally, since performance may vary depending on window size, we display the average AUROC across all window sizes that we have considered in our experiments. These window sizes include: $\{10, 20, 50, 100, 200, 500, 1000\}$. In Figure 4, we summarize our findings by displaying the average AUROC value as a function of the chosen hyper-parameters. These results are presented as heatmaps.

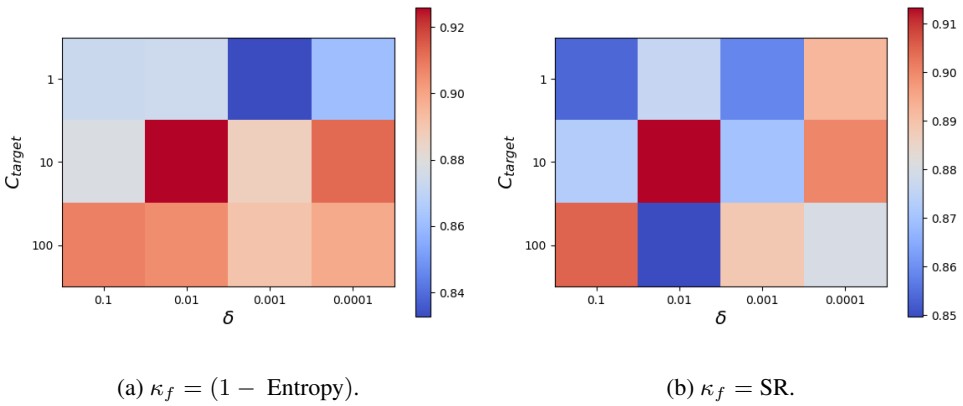

(a) $\kappa_f = (1 - \text{Entropy})$.      (b) $\kappa_f = \text{SR}$.

Figure 4: AUROC performance of our detector under different choices of hyper-parameters.

Figure 4a, displays our AUROC detector's performance when we use Entropy-based as our CF. We observe that the optimal choice of hyper-parameters is $\delta = 0.01$ and $C_{\text{target}} = 10$, resulting in the highest performance. However, increasing the value of $C_{\text{target}}$ leads to a more consistent and robust detector, as changes in the value of $\delta$ do not significantly affect the detector's performance. Additionally, we note that using $C_{\text{target}} = 1$ yields relatively poor performance, indicating that a single coverage choice is insufficient to capture the characteristics of the distribution represented by the sample $S_m$. Similar results are obtained when using SR as the CF, as shown in Figure 4b. These results suggest that selecting a high value of $C_{\text{target}}$ and a low value of $\delta$ is the most effective approach for ensuring a robust detector. Finally, the heatmaps demonstrate that Entropy-based CF outperforms (by a low margin) SR, in terms of detection performance.

# F Upper Coverage Bound Development and Application

In this section, we discuss the complementary problem to that outlined in Section 4.1, which is solved by Algorithm 3; we also show its application for detecting distribution shifts, Algorithm 4. Finally, we introduce a unified algorithm, Algorithm 5, for detecting distribution shifts, while accounting for both *under-coverage* and *over-coverage*, details follow.

**Problem statement.** *For a classifier $f$, a detection-training sample $S_m \sim \mathcal{P}^m$, a confidence parameter $\delta > 0$, and a desired coverage $c^* > 0$, our goal is to use $S_m$ to find a $\theta$ value (which implies a selection function $g_\theta$), such that the coverage satisfies,*

$$\mathbf{Pr}_{S_m}\{c(\theta, \mathcal{P}) > c^*\} < \delta. \tag{6}$$

This means that *over coverage* should occur with probability of at most $\delta$.

The pseudo code of the algorithm that finds the optimal coverage upper bound (with confidence $\delta$) – SGC-UP, appears in Algorithm 3.

---

**Algorithm 3:** *Selection with guaranteed coverage - Upper bound (SGC-UP)*

---

1   **Input:** train set: $S_m$, confidence-rate function: $\kappa_f$, confidence parameter $\delta$, target coverage: $c^*$.
2   Sort $S_m$ according to $\kappa_f(x_i)$, $x_i \in S_m$ (and now assume w.l.o.g. that indices reflect this ordering).
3   $z_{\min} = 1$, $z_{\max} = m$
4   **for** $i = 1$ **to** $k = \lceil \log_2 m \rceil$ **do**
5      $z = \lceil (z_{\min} + z_{\max})/2 \rceil$
6      $\theta_i = \kappa_f(x_z)$
7      Calculate $\hat{c}_i(\theta_i, S_m)$
8      Solve for $b_i^*(m, m \cdot \hat{c}_i(\theta_i, S_m), \frac{\delta}{k})$ {see Lemma F.1}
9      **if** $b_i^*(m, m \cdot \hat{c}_i(\theta_i, S_m), \frac{\delta}{k}) \geq c^*$ **then**
10        $z_{\min} = z$
11      **else**
12        $z_{\max} = z$
13      **end if**
14   **end for**
15   **Output:** bound: $b_k^*(m, m \cdot \hat{c}_k(\theta_k, S_m), \frac{\delta}{k})$, threshold: $\theta_k$.

---

To deduce Equation (6) we make use of Lemma F.1, which is nearly identical to Lemma 4.1.

**Lemma F.1.** *Let $\mathcal{P}$ be any distribution and consider a CF threshold $\theta$ with a coverage of $c(\theta, \mathcal{P})$. Let $0 < \delta < 1$ be given and let $\hat{c}(\theta, S_m)$ be the empirical coverage w.r.t. the set $S_m$, sampled i.i.d. from $\mathcal{P}$. Let $b^*(m, m \cdot \hat{c}(\theta, S_m), \delta)$ be the solution of the following equation:*

$$\underset{b}{arg\,min} \left( \sum_{j=0}^{m \cdot \hat{c}(\theta, S_m)} \binom{m}{j} b^j (1-b)^{m-j} \leq \delta \right). \tag{7}$$

*Then,*

$$\mathbf{Pr}_{S_m}\{c(\theta, \mathcal{P}) > b^*(m, \hat{c}(\theta, S_m), \delta)\} < \delta. \tag{8}$$

The following is a uniform convergence theorem for SGC-UP (Algorithm 3) procedure stating that all the calculated bounds are valid simultaneously with a probability of at least $1 - \delta$.

**Theorem F.2.** *Assume $S_m$ is sampled i.i.d. from $\mathcal{P}$, and consider an application of Algorithm 3. For $k = \lceil \log_2 m \rceil$, let $b_i^*(m, m \cdot \hat{c}_i(\theta_i, S_m), \frac{\delta}{k})$ and $\theta_i$ be the values obtained in the $i^{th}$ iteration of Algorithm 3. Then,*

$$\mathbf{Pr}_{S_m}\{\exists i : c(\theta_i, \mathcal{P}) > b_i^*(m, m \cdot \hat{c}_i(\theta_i, S_m), \frac{\delta}{k})\} < \delta.$$

The proof for Theorem F.2 can be easily deduced from the proof of Theorem 4.2, given in Appendix B.1.

## F.1   Application of Upper Bound Algorithm for Detecting Distribution Shifts

In this section, we demonstrate the application of Algorithm 3 for detecting shifts in distribution. Specifically, Algorithm 3 is useful in identifying distribution shifts towards 'over-confidence,' where the model exhibits excessive confidence in its predictions on input data. It's important to note that this scenario was not empirically observed in our experiments, leading to the exclusion of this section from the main body of the paper.

Our 'over-confidence' detection algorithm is analogous to Algorithm 2. We start by applying SGC-UP to $C_{\text{target}}$ target coverages uniformly spread between the interval $[0.1, 1]$, excluding the coverage of 1. Each application $j$ of SGC-UP on the same sample $S_m \sim \mathcal{P}_m$ with a target coverage of $c_j^{*;\text{up}}$ produces a pair: $(b_j^{*;\text{up}}, \theta_j^{\text{up}})$, which represents a threshold and a bound, respectively.

Define, $\delta^{\text{up}}(\hat{c}|b^*) = \mathbb{1}[\hat{c} \geq b^*]$, thus, given a window of k samples from an alternative distribution, $W_k \sim Q^k$, we define the *sum of upper bound violations*, $V^{\text{up}}$, as follows,

$$
\begin{aligned}
V^{\text{up}} &= \frac{1}{C_{\text{target}}} \sum_{j=1}^{C_{\text{target}}} (\hat{c}(\theta_j^{\text{up}}, W_k) - b_j^{*;\text{up}}) \cdot \delta^{\text{up}}(\hat{c}(\theta_j^{\text{up}}, W_k)|b_j^{*;\text{up}}) \\
&= \frac{1}{k \cdot C_{\text{target}}} \sum_{j=1}^{C_{\text{target}}} \sum_{i=1}^{k} (g_{\theta_j^{\text{up}}}(x_i) - b_j^{*;\text{up}}) \cdot \delta^{\text{up}}(\hat{c}_j^{\text{up}}|b_j^{*;\text{up}}),
\end{aligned}
\tag{9}
$$

where we obtain the last equality by using $\hat{c}_j^{\text{up}} \triangleq \hat{c}(\theta_j^{\text{up}}, W_k) = \frac{1}{k} \sum_{i=1}^{k} g_{\theta_j^{\text{up}}}(x_i)$.

Similarly to $V$ in Equation (4), the term $V^{\text{up}}$ denotes the magnitude of the violation, where $V^{\text{up}} = 0$ indicates an absence of over-confidence in the predictions.

A pseudocode of our algorithm for detecting over-confidence in predictions appears in Algorithm 4.

---

**Algorithm 4:** *Coverage-Based Detection - over confidence*

---

1   // Fit
2   **Input Training:** $S_m, \delta, \kappa_f, C_{\text{target}}$
3   Generate $C_{\text{target}}$ uniformly spread coverages $\boldsymbol{c}^*$
4   **for** $j = 1$ **to** $C_{target}$ **do**
5     $b_j^{*;\text{up}}, \theta_j^{\text{up}} = \texttt{SGC-UP}(S_m, \delta, c_j^*, \kappa_f)$
6   **end for**
7   **Output Training:** $\{(b_j^{*;\text{up}}, \theta_j^{\text{up}})\}_{j=1}^{C_{\text{target}}}$
8   // Detect
9   **Input Detection:** $\{(b_j^{*;\text{up}}, \theta_j^{\text{up}})\}_{j=1}^{C_{\text{target}}}, \kappa_f, \alpha, k$
10   **while** True **do**
11     Receive window $W_k = \{x_1, x_2, \ldots, x_k\}$
12     Calculate $V^{\text{up}}$ {see Equation (9)}
13     Obtain p-value from t-test, $H_0 : V^{\text{up}} = 0, H_1 : V^{\text{up}} > 0$
14     **if** $p_{value} < \alpha$ **then**
15       Shift_detected $\leftarrow$ True
16       **Output Detection:** Shift_detected, $p_{\text{value}}$
17     **end if**
18   **end while**

---

### F.2   Unified Algorithm for Distribution Shift Detection

In this section, we introduce a unified algorithm designed to detect shifts in distribution, accommodating both scenarios of over-confidence and under-confidence in predictions.

Define *sum of violations* (SOV),

$$
\text{SOV} \triangleq V + V^{\text{up}} = \frac{1}{C_{\text{target}}} \sum_{j=1}^{C_{\text{target}}} (\hat{c}_j^{\text{up}} - b_j^{*;\text{up}}) \cdot \delta^{\text{up}}(\hat{c}_j^{\text{up}}|b_j^{*;\text{up}}) + (b_j^* - \hat{c}_j) \cdot \delta(\hat{c}_j|b_j^*).
\tag{10}
$$

A pseudocode of our unified algorithm for detecting both over and under confidence in predictions appears in Algorithm 5.

**Algorithm 5:** *Coverage-Based Detection - Unified*

1    // Fit
2    **Input Training:** $S_m, \delta, \kappa_f, C_{\text{target}}$
3    Generate $C_{\text{target}}$ uniformly spread coverages $\boldsymbol{c}^*$
4    **for** $j = 1$ **to** $C_{target}$ **do**
5       $b_j^*, \theta_j = \text{SGC}(S_m, \delta, c_j^*, \kappa_f)$
6       $b_j^{*;\text{up}}, \theta_j^{\text{up}} = \text{SGC-UP}(S_m, \delta, c_j^*, \kappa_f)$
7    **end for**
8    **Output Training:** $\{(b_j^*, \theta_j)\}_{j=1}^{C_{\text{target}}}, \{(b_j^{*;\text{up}}, \theta_j^{\text{up}})\}_{j=1}^{C_{\text{target}}}$
9    // Detect
10    **Input Detection:** $\{(b_j^*, \theta_j)\}_{j=1}^{C_{\text{target}}}, \{(b_j^{*;\text{up}}, \theta_j^{\text{up}})\}_{j=1}^{C_{\text{target}}}, \kappa_f, \alpha, k$
11    **while** True **do**
12       Receive window $W_k = \{x_1, x_2, \ldots, x_k\}$
13       Calculate SOV {see Equation (10)}
14       Obtain p-value from t-test, $H_0 : \text{SOV} = 0, H_1 : \text{SOV} > 0$
15       **if** $p_{value} < \alpha$ **then**
16         Shift_detected $\leftarrow$ True
17         **Output Detection:** Shift_detected, $p_{\text{value}}$
18       **end if**
19    **end while**

