# OpenReview forum: "Window-Based Distribution Shift Detection for Deep Neural Networks"
_NeurIPS.cc/2023/Conference — NeurIPS 2023 poster_

### Official Review · Reviewer_UhJJ · 2023-06-29

**Soundness:** 3 good
**Presentation:** 4 excellent
**Contribution:** 3 good
**Rating:** 7
**Confidence:** 3

**Summary:**

This paper studies the topic of model monitoring through the lenses of input distribution shift detection during test time. They propose a distribution deviation detection method based on selective prediction coverage bound violation counting. They detect data distribution shift based on windows and studies the adaptation of single-instance confidence functions for this task. They compare to baselines such as the KS test and MMD method over the softmax or penultimate layer embeddings, and two single instances baselines based on the maximum softmax response and the output entropy with a t-test. The experimental benchmark is over ImageNet with popular models and shifts based on adversarial attacks, data corruption, and semantically different data.

**Strengths:**

- They present an original perspective on drift detection inspired from selective classification focused on real-world application.

- The algorithm exhibits great time complexity, space complexity, and good performance.

- Clean theoretical formulation of their algorithm, easy to understand.

- Extensive experimental benchmark and comprehensive ablation study on the hyperparameters.

- Overall, the paper is well-written and easy to follow.

**Weaknesses:**

- The choice of confidence-rate function might be a bottleneck of the method. Some models can exhibit over-confident predictions even to far OOD data. Relying on the softmax output as a confidence method might present a risk.

- The formulation by counting the violations of the coverage bounds is interesting but it lacks motivation. It is hard to understand why computing the coverage bound would be more powerful than the t-tests with the confidence function values for each $x_k$.

- Contribution (1) does not seem very useful. Why would someone desire to limit coverage without controlling risk?

- In Table 1, the time complexities for MMD and KS might be inflated, as they could be improved by slightly changing the implementation. E.g., for MMD, the term with $O(dm^2)$ time complexity could be computed offline on a fit step.

### Secondary observations

In line 375 the authors defined FPR@95TPR but following on in the paper they call it TNR@95TPR. Figure 4 would be easier to read if the AUROC numbers were written in the heatmap instead of the colorbars.

**Questions:**


1. The test set evaluation setup is unclear to me. The testing windows are pure out-of-distribution/corrupted data? What about mixed windows with in-distribution and ood data, which would be a more realistic setting. Have the author tried windows with a mixture of in-distribution and out-of-distribution data? I would strongly encourage adding an experiment in these lines to strengthen the experimental part of the paper.

2. For the case of k=1000, is there a single window? It would mean that you compute the prediction for each method over the 15 seeds and compute the metrics in these predictions?

3. Could the authors elaborate more on the claim that "detection power might lay within lower coverages" (line 362)? To my understanding, lower coverage simply gives a higher (more selective) threshold $\theta$ for the CF, thus, it rejects more less confident samples.

4. Could you explain in more details the novelty of Algorithm 1, Lemma 4.1 and Theorem 4.2 w.r.t references [11,12]?

5. A more open ended less important question. Why do some baselines and your method decrease/oscillate performance with window size on some types of drift?


**Limitations:**

The authors do not discuss the limitations of their work in great detail. A greater effort in this direction would be appreciated. For instance, on ImageNet-O, there appears that MMD Embeddings exibits better performance. So maybe the presented method is more appropriated to some types of drifts. Also, I would argue that a variation of 5% in AUROC on the ablation of $\delta$ and $C_{target}$ shows some sensitivity to hyperparameters.

---

> ### Author Rebuttal · Authors · 2023-08-07
>
> Thank you for the positive feedback! we'll address below the issues raised.
>
> **"The choice of confidence-rate function might be.."**
>
> We understand your point, and in addition to Algorithm 1, we have developed a symmetric algorithm that construct an upper coverage bound (as opposed to a lower coverage bound). This algorithm can be valuable in detecting overconfidence in predictions. Although we did experiment with this bound, we found it to be rarely useful in practice, and thus decided not to include it in the paper. However, to address your concern and also for the sake of completeness, we will include an Appendix section that introduces the symmetric algorithm and provides a discussion in response to your point.
>
> **"The formulation by counting the violations of the coverage.."**
>
> The motivation behind using the coverage bound stems from a complexity standpoint. By employing the bound, we can encode all essential information for detecting distribution shifts using only $C_{target}$ scalars, which in our experiments is set to $C_{target}=10$. In contrast, the 2-sample t-test would require the data from both the in-distribution (typically very large) and the window data to make a detection, making it (significantly) less efficient.
>
> **"Contribution (1) does not seem very useful. Why would.."**
>
> You raise an interesting/important point. We will try to motivate this algorithm with a practical scenario. Consider a hospital where doctors have the capacity to examine only 70\% of the available X-ray results by the next day, yet all results must be analyzed within this time-frame. Our algorithm can help in this situation by constructing a selective classifier. This classifier is assured (with a probability of $1-\delta$) to cover the remaining 30\% of the X-rays that the doctors are unable to reach. Importantly, our selective classifier ensures that this 30\% is the portion upon which it would have the least risk.
>
> **"In Table 1, the time complexities for MMD and KS might be.."**
>
> We see your point. Indeed, the term with $O(dm^2)$ time complexity in MMD can be computed offline during a fit step and we'll clarify this in the final version of the paper. However, we would like to point out that the remaining terms in MMD ($mk + k^2$), along with the complexity of the KS algorithm, cannot be further improved, considering they are computed based on the interaction between in-distribution (ID) and potential OOD data. In simpler terms, unlike our method whose detection complexity remains independent of the ID size ($m$), the baselines still exhibit dependence on the ID size ($m$), rendering them unsuitable for handling ``Google-Scale'' datasets.
>
> **"In line 375 the authors defined.."**
>
> Thanks! Indeed we mean ``FPR@95TPR'', this will be fixed for the final version.
>
> **"Figure 4 would be easier to.."**
>
> That's a good point. We will change this for the final version.
>
> **"The test set evaluation setup is unclear to me.."**
>
> To clarify, each window exclusively contains OOD samples, not a mix of both. Nevertheless, to directly address your point, we draw your attention to a preliminary experiment, that we have conducted in the beginning stages of this work. In this experiment, we deliberately adjusted the proportion ratio $p$ to allow a mix of OOD and in-distribution (ID) samples within a window. The ImageNet dataset was used as ID and EfficientNetB0 was chosen as the model. For illustration, we present in Table 3 in the attached PDF file, the results corresponding to a ratio of $p=1/3$, i.e., one third of the window is contaminated with OOD data. We will be happy to include these preliminary results in the final version of the paper if the reviewer thinks that this is important.
>
> **"For the case of k=1000.."**
>
> Yes.
>
> **"Could the authors elaborate more.."**
>
> You are right about this claim. To further clarify, we direct your attention to Fig. 2(b) in the paper. In this figure, you can observe the margin between the violations (indicated by the red circles) and the coverage bounds (indicated by the black line). Notably, the largest margin (violation) occurs when the target coverage is around 0.4. This implies that the violations (which are indicators of a distribution shift) are more substantial when using lower coverages (lower than 1); thus, we suggested that the ``detection power might lay within lower coverages''.
>
> **"Could you explain in more details.."**
>
> Certainly,  our Algorithm 1, Lemma 4.1, and Theorem 4.2 function without relying on labeled data; by focusing on coverage, rather than risk. This significantly contrasts with the approaches outlined in ref. [11,12], as their reliance on labeled data disqualifies them as engines for detecting distribution shifts. Thus, although Algorithm 1 seems similar to those found in the references mentioned, they are semantically completely different in the sense that they are designed for completely different use cases. By focusing on coverage, our technique offers a unique feature that isn’t found in [11, 12].
>
> **"A more open ended less important.."**
>
> We hypothesize that it stems from the number of random splits used to average the results. To note, we've utilized 15 random splits in our study, a significant number when taking into account common practice in distribution shift detection. For comparison, the study referenced as [10] (in our paper) used 5 random splits.
>
> **"The authors do not discuss the.."**
>
> We appreciate this comment. In response, we will enhance Appendix D with added insights. Moreover, in relation to your concerns about limitations, please refer to lines 33-41 in the paper, where we discuss the limitations of our work. To clarify, the primary limitation of our study is that we don't consider identifying single-instance out-of-distribution examples, characterizing types of distribution shifts, or addressing out-of-distribution robustness. If the reviewer advises, we'll further emphasize these limitations in the final version.

---

> > ### Comment · Reviewer_UhJJ · 2023-08-14
> >
> > I thank the authors for all the effort put in the rebuttal and for the additional experiments.
> >
> > 1. I believe the two-sided version of the algorithm is an interesting theoretical contribution as well, even though less practical. Thus I suggest adding it to a section in the appendix as you have already studied it, and it might be interesting in the future with other types of drift or attacks.
> > 2. Thank you for elaborating on this example to motivate Contribution (1). Having it early in the paper might help the reader better understand in which direction you are going.
> > 3. The experiment with a mixed window of ID and OOD data is important, as it reflects better challenging application scenarios. I'm glad that you ran experiments with it.
> > 4. Thank you for clarifying certain aspects w.r.t references [11,12].
> > 5. This might be just a question of writing style, but I suggest the authors to re-work the paragraph in which they segment their objective and the limitations into two separate paragraphs (L.26-41). I reckon it contains too much information.
> >
> > All of my concerns were addressed satisfactorily. With the richer experimental section and better motivation, I increase my recommendation for acceptance.

---

### Official Review · Reviewer_M7ue · 2023-07-06

**Soundness:** 3 good
**Presentation:** 2 fair
**Contribution:** 2 fair
**Rating:** 6
**Confidence:** 2

**Summary:**

The paper proposes a window-based distribution shift detection method to detect whether the samples in the window are in-distribution (ID) or OOD. The method takes a window containing samples as input and computes a confidence score using the proposed coverage-based detection algorithm. The method is evaluated on various OOD benchmarks when imagenet is ID dataset. The results demonstrate that the proposed method is computationally more efficient than the existing methods and achieve high accuracies in majority cases.

**Strengths:**

- The idea of detecting OOD samples within a window is interesting and in a different direction than the majority of OOD detection literature.
- The paper aims to present a theoretically supported and computationally efficient algorithm for this problem.

**Weaknesses:**

- Lack of intuition and convoluted description of the method: The intuition behind the proposed method is unclear. I didn't quite understand why this method should perform better than the existing ones. The theoretical part should be explained after giving some motivation for the method.

- Lack of experiments: The paper presents results only when ImageNet is ID dataset. There should be experiments with different ID dataset to understand if the results generalize to other datasets.

- Some experimental details are not clear: It is unclear to me how windows are created in test time. Can a window contain both ID and OOD samples? Since the method labels the whole window as OOD or ID, I wonder how the method's performance changes with different ID and OOD sample ratios within a window.

**Questions:**

Please address my concerns in the weaknesses section.

**Limitations:**

Limitations of the method are not discussed.
I don't see any potential negative societal impact of the work.

---

> ### Author Rebuttal · Authors · 2023-08-06
>
> We appreciate your feedback. We have provided detailed point-to-point responses, which we believe can address your major concerns. We wonder whether you would kindly reconsider your rating based on our responses to your questions/concerns. Your consideration is highly appreciated. Many thanks!
>
> **"Lack of intuition and convoluted description of the method: The intuition behind the proposed method is unclear. I didn't quite understand why this method should perform better than the existing ones. The theoretical part should be explained after giving some motivation for the method."**
>
> We understand your concerns and thank you for pointing out areas for improvement. The essence of our method's intuition is illustrated in Figure 2,b (p.8, Section 6.2). The coverage violation peaks when its value approaches 0.4, thus providing a compelling rationale for adopting coverages under 1. Unlike our approach, the baselines do not account for coverages less than 1, and we demonstrate (in Table 2) that lower coverages strengthen distribution shift detection. As for the theoretical part, in light of your comments, we will expand our theoretical discussion in the final version of the paper. Specifically, we intend to introduce a clarifying paragraph at the beginning of sections 4.1 and 4.2 to further clarify the theoretical aspects of our method. In any case, let us explain our method in a few words: we develop a robust generalization coverage bound that holds with high probability for data sampled from the source distribution. The main idea is that if this coverage bound is violated, it likely signifies a shift in the distribution.
>
> **"Lack of experiments: The paper presents results only when ImageNet is ID dataset. There should be experiments with different ID dataset to understand if the results generalize to other datasets."**
>
> We understand your point. While we are unable to conduct a new experiment within the rebuttal time frame, we can illustrate the generalizability of our method to other datasets by presenting results from an experiment that we have conducted (in early stages of the work presented in the paper) on the CIFAR10 dataset (as the ID dataset). This experiment was done under a nearly identical setting, but we have not included it in the paper. We used two pre-trained models, ResNet-18 and ResNet-50, and implemented the following simulated shifts:
>
> * **Rotations**: with angles $\theta \in$ \{$10, 40, 90$ \}.
> * **Gaussian noise**: with STD $\sigma \in$ \{ $\frac{40}{255}, \frac{60}{255}, \frac{80}{255}$ \}.
> * **OOD Datasets**: employing the CIFAR-100 dataset and the SVHN dataset as OOD datasets.
> * **Adversarial**: fast gradient sign method, projected gradient descent, and Carlini Wagner, abbreviated as FGSM, PGD, and CW respectively.
>
> The results of this experiment can be found in Table 1 in the attached PDF file. If the reviewer suggests it, we are prepared to incorporate those results into the final paper.
>
> **"Some experimental details are not clear: It is unclear to me how windows are created in test time. Can a window contain both ID and OOD samples? Since the method labels the whole window as OOD or ID, I wonder how the method's performance changes with different ID and OOD sample ratios within a window."**
>
> To clarify this point, each window exclusively contains OOD samples, not a mix of both. Nevertheless, to directly address your point, we would like to present a preliminary experiment, which again, was conducted in the beginning stages of this work (and was not included in the paper). In this experiment, we deliberately adjusted the proportion ratio $p$ to allow a mix of OOD and ID samples within a window. The ImageNet dataset was used as ID and EfficientNetB0 was chosen as the model. As an example, we show in Table 3 in the attached PDF file, the results corresponding to a ratio of $p=1/3$, i.e., one-third of the window is contaminated with OOD data.
>
> Again, we will be happy to include those results in the final paper.
>
> **"Limitations of the method are not discussed."**
>
> We want to refer the reviewer to lines 33-41, where we discuss the limitation of our work. Notably, we do not address single-instance OOD detection, nor do we characterize the nature of distribution shifts. To further emphasize that, we will broaden this discussion for the final version of the paper.

---

> > ### Comment · Reviewer_M7ue · 2023-08-16
> >
> > Thanks to the authors for providing a detailed rebuttal to all reviewers and engaging in the discussion. I read all the other reviewers' comments and the author's responses. I do not have any further concerns about the paper and support its acceptance. I suggest authors revise the paper considering the reviewers' comments.

---

### Official Review · Reviewer_7FwS · 2023-07-06

**Soundness:** 3 good
**Presentation:** 3 good
**Contribution:** 3 good
**Rating:** 6
**Confidence:** 3

**Summary:**

The paper proposes a methodology to detect distribution shifts in windows (set) of observations for pre-trained deep learning models on computer vision tasks. The method compares the empirical coverage of a selection function to pre-determined computed bounds. During the fit stage, a guaranteed coverage selection algorithm determines C_target different bounds. During inference, the method computes the empirical coverage of the window to compute the total violations to the bounds. Finally, since V follows a normal distribution, a t-test statistic detects distribution shifts. The proposed method requires orders of magnitude less time and memory than the current SotA for large datasets. The paper presents a comprehensive comparison with SoTA methods in ImageNet.


**Strengths:**

- The paper focuses on identifying distribution shifts, a very important problem with multiple applications.
- The methodology is appealing and very simple.
- The proposed method is theoretically supported.
- The results demonstrate the method is faster and requires less memory than the current SoTA.
- The paper is well written, and the ideas are clearly motivated.
- The experimental setting is comprehensive and well-designed.
- Code to replicate results is provided.

**Weaknesses:**

- The proposed method performs better on smaller windows (k~50), which is contrary to the main claims of the paper that the proposed method can leverage population-level information.
- One of the main applications to motivate the problem is determining the need to retrain pre-trained models. However, the size of windows (k) used in the experiments seems small for this application, especially since large models are potentially deployed in datasets with millions of observations.
- The performance benefits of larger samples (k~1,000) are smaller than existing methods such as KS.
- The paper only tests the simple method where V is an average of the C_target scalars.
- Coverage is an established metric, and it could be directly used as a distribution-shift detector for the type of application discussed in the paper. It would be interesting to compare the method with a single coverage target value to better understand the benefits of the extension proposed in the paper. Appendix E presents this ablation study but is missing some important information. For instance, what value of C^* is used when C_target=1?
- Limitations are not properly discussed. It seems that the accuracy advantages of the method are limited to the less useful setting where k is small.

**Questions:**

- Which target coverage is used in the appendix when C_target=1?
- Have authors considered using other datasets than ImageNet?

**Limitations:**

The paper does not properly discuss limitations (see above).

---

> ### Author Rebuttal · Authors · 2023-08-07
>
> Thank you for the positive feedback. We are glad that you found the paper well-written and appreciated the theoretical and experimental analysis. We will try to address each of the issues you raised.
>
> **"The proposed method performs better on smaller windows (k~50), which is contrary to the main claims of the paper that the proposed method can leverage population-level information."**
>
> We see your concern. Please note that by ``population-based'', we intended to refer to any case where $k > 1$; since the literature is basically divided to methods applicable for $k=1$ (for example references [1,24,25]), and for $k>1$ (for example references [9, 10]). We will clarify this in the final version.
>
> **"One of the main applications to motivate the problem is determining the need to retrain pre-trained models. However, the size of windows (k) used in the experiments seems small for this application, especially since large models are potentially deployed in datasets with millions of observations."**
>
> We wish to clarify that the choice of $k$ as a maximum value of $1,000$ was guided by our empirical findings that most metrics get a nearly perfect score around this point (for the top-performing methods). For example, our method exhibits a score of 99+ for the metrics AUROC, AUPR-In and AUPR-Out for $k=1000$, for all architectures considered; see Table 2 in the paper. Furthermore, the detectors' performance can be observed to increase with $k$, a trend illustrated in Table 2, therefore, larger values of $k$ will just strengthen the scores.
>
> **"Coverage is an established metric, and it could be directly used as a distribution-shift detector for the type of application discussed in the paper. It would be interesting to compare the method with a single coverage target value to better understand the benefits of the extension proposed in the paper. Appendix E presents this ablation study but is missing some important information. For instance, what value of $C^\ast$ is used when $C_{target}=1$?"**
>
> We see your point. In the scenario you mentioned, where $C_{target}=1$, we have set $C^* = 0.5$. We recognize that this information is missing from Appendix E, and we will ensure to include it in the revised version of the paper. If there are any other details or clarifications needed, we will be happy to provide them.
>
> **"Limitations are not properly discussed. It seems that the accuracy advantages of the method are limited to the less useful setting where k is small."**
>
> We refer the reviewer to the second paragraph of the introduction section, specifically lines 33-41, where we discuss the limitations of our work. To clarify, our study's primary limitation is that we don't consider identifying single-instance out-of-distribution examples, characterizing types of distribution shifts, or addressing the complementary topic of out-of-distribution robustness. Regarding your comment about the value of $k$, we think that superior performance on smaller $k$ values is actually more useful; since in our observations, most metrics get a nearly perfect score when $k=1000$ (for the top-performing methods). For example, our method exhibits a score of 99+ for the metrics AUROC, AUPR-In and AUPR-Out for $k=1000$, for all architectures considered; see Table 2 in the paper.
>
> **"Have authors considered using other datasets than ImageNet?"**
>
> Yes. While the time constraint for rebuttal prevents us from conducting new experiments, we can indeed share some findings from an experiment under nearly identical setting, conducted at the beginning stages of the paper. We used CIFAR10 as the ID dataset, and employed two pretrained models, ResNet-18 and ResNet-50. We simulated the following shifts:
> * **Rotations**: We implemented rotations with angles $\theta \in ${ $10, 40, 90$ }.
> * **Gaussian noise**: We introduced Gaussian noise with standard deviations $\sigma \in$ { $\frac{40}{255}, \frac{60}{255}, \frac{80}{255}$}.
> * **OOD Datasets**: We utilized the CIFAR-100 and SVHN datasets as out-of-distribution (OOD) sets.
> * **Adversarial**: We employed the fast gradient sign method, projected gradient descent, and Carlini Wagner, commonly referred to as FGSM, PGD, and CW respectively.
>
> The results can be found in Table 1 in the attached PDF file. Should the reviewer recommend it, we will be happy to include those results in the final paper.
>
> **References**
>
> [1] Enhancing the reliability of out-of-distribution image detection in neural networks. ICLR.
>
> [9] Detecting and correcting for label shift with black box predictors. ICML.
>
> [10] Failing loudly: An empirical study of methods for detecting dataset shift. NeurIPS.
>
> [24] Detecting out-of-distribution examples with gram matrices.ICML.
>
> [25] A simple unified framework for detecting out-of-distribution samples and adversarial attacks. NeurIPS.

---

### Official Review · Reviewer_BTJN · 2023-07-14

**Soundness:** 3 good
**Presentation:** 3 good
**Contribution:** 2 fair
**Rating:** 5
**Confidence:** 3

**Summary:**

The paper presents a new method for detecting distribution shifts within a given window of samples. This coverage-based detection algorithm is theoretically motivated and can be applied to pre-trained models. The authors claim the low computational complexity of their framework, and experiments show that the framework outperforms the baselines across a variety of distribution shifts such as pixel-space adversarial examples.

**Strengths:**

1. The paper is well-written with a good storyline that formulates the proposed framework.
2. The window setting is an attractive problem that has yet to be well explored by past works. The target models comprehensively include convolution-based and attention-based.
3. The paper well defines the domain and assumptions of the task, making it clearly comparable to OOD frameworks.
3. The authors motivate the framework by theoretical analysis, as well as conducting empirical experiments that demonstrate the effectiveness of the framework.


**Weaknesses:**

1. From the perspective of pixel-space adversarial attack detection, authors only evaluate the most basic baselines of FGSM and PGD. It will be constructive to the validation of the work if authors conduct additional experiments on the CW attack [1] and GAN perturbations [2].

2. Authors intend to claim that the framework is effective to detect naturally (semantically) mutated images. However, the authors did not elaborate sufficient validations on the line of research for semantic adversarial examples [3]. It will be beneficial to the work if the authors have discussions on the effectiveness of the work for this line of research.

3. The framework should be evaluated on scalable benchmarks proposed in recent years such as WILDS [4] V1/V2.

[1] Towards Evaluating the Robustness of Neural Networks

[2] Generating Adversarial Examples with Adversarial Networks

[3] Semantic Adversarial Attacks: Parametric Transformations That Fool Deep Classifiers.

[4] WILDS: A Benchmark of in-the-Wild Distribution Shifts

**Questions:**

Please address my concerns and questions in the weakness section.

To summarize, this paper is a fair contribution to the OOD research. Although the paper is well-structured with a clear storyline, given several major concerns stated in the weakness section, I will rate it as borderline accept at this stage. I will keep track of authors' responses, and consider revising the rating based on the soundness of responses.

**Limitations:**

The authors discussed potential improvements and future works in section 7.

---

> ### Author Rebuttal · Authors · 2023-08-06
>
> We're pleased to note from your feedback that you recognize the importance of our research topic and believe our paper is a valuable addition to OOD research. We will try to address each of the issues you raised. We hope, that based on our responses, you would kindly reconsider your rating. Your consideration is highly appreciated!
>
> **"1) From the perspective of pixel-space adversarial attack detection, authors only evaluate the most basic baselines of FGSM and PGD. It will be constructive to the validation of the work if authors conduct additional experiments on the CW attack [1] and GAN perturbations [2].
> 2) Authors intend to claim that the framework is effective to detect naturally (semantically) mutated images. However, the authors did not elaborate sufficient validations on the line of research for semantic adversarial examples [3]. It will be beneficial to the work if the authors have discussions on the effectiveness of the work for this line of research."**
>
> 1)+2) We appreciate the reviewer's suggestion to incorporate additional evaluation using CW attack [1] and GAN perturbations [2], and we indeed would like to include that in the paper. While it may not be feasible to run large-scale experiments within the rebuttal time frame, we would like to present a preliminary experiment we have conducted using EfficientNetB0 as the model, ImageNet as the ID (in distribution) dataset, and to your request, the CW attack as the OOD (out of distribution) dataset. These preliminary results are presented in Table 2 in the attached PDF file. In response to your request, we will be happy to include these results, extend upon them, and further discuss them in the final paper.
>
> **"3) The framework should be evaluated on scalable benchmarks proposed in recent years such as WILDS [4] V1/V2."**
>
> Although we believe we have conveyed the strength of our method across a wide range of shifts -- such as ImageNet-O ([40]), ImageNet-A ([40]), adversarial ([38,39]), rotations, and more -- we acknowledge that this would be a nice addition to our work. Thus, should time allow, we will do our best to incorporate some results along this line in the final version of the paper. We would like to note, though, that our methodology was inspired by reference [10] and expanded upon it, both in terms of the OOD datasets employed and the variety of models utilized.
>
> **References**
>
> [10]  Failing loudly: An empirical study of methods for detecting dataset shift. NeurIPS,
>
> [38] Explaining and harnessing adversarial examples. ICLR.
>
> [39] Towards deep learning models resistant to adversarial attacks. ICLR.
>
> [40] Natural adversarial examples. CVPR.

---

> > ### Comment · Reviewer_BTJN · 2023-08-11
> >
> > Thanks for providing a rebuttal with fair arguments. I will keep the current rating.

---

### Official Review · Reviewer_HyPc · 2023-07-17

**Soundness:** 2 fair
**Presentation:** 2 fair
**Contribution:** 3 good
**Rating:** 5
**Confidence:** 3

**Summary:**

A method to detect shifts in the input distribution of deep neural-networks is presented. The method takes a window of samples as input and calculates deviation from a tight coverage generalization bound. Here, coverage is empirically defined as the fraction of input samples that are non-rejected based on a confidence function, i.e. those which are predicted with sufficiently high confidence. The method is evaluated on standard image classification datasets using various commonly used CNN architectures.

**Strengths:**

The method is well motivated and the detection metric ('sum bound violations') is defined and derived in a principled manner. Since the method is based on principles of selective classification, a by-product is a confidence threshold $\theta$ and a corresponding selection function $g_\theta(x)$, which are useful to have in practical situations. What is interesting is that the method does not detection a distribution-shift based on uncertainty or confidence of the input samples, but rather by the deviation of coverage from pre-calculated bounds at several target values based on these uncertainties. Similar to reference [12], the method does not rely on uncertainty values being calibrated.

**Weaknesses:**

1. The derivation of the main steps of the algorithm (Section 4.1) mirrors closely the derivation of the selection-based classifier in Section 3 of the paper cited as reference [12] in the manuscript. This isn't a weakness or limitation in my view, but it is worth noting the close conceptual similarities.

2. What is concerning though is the rather weak experimental evaluation:

    a. The authors claim that single-instance will not perform as well as their window-based method. However, the only single-instance baselines compared against are softmax response and entropy. These are rather weak baselines and there are several stronger baselines (several of which have been cited by the authors) that should have been compared against.

    b. It would have been really interesting and useful to see this method applied with different types of uncertainty values apart from softmax (feature-based such as Mahalanobis, predictive uncertainty from Bayesian neural networks, etc.). That would have served as a much stronger evidence of its superiority over other methods such as KS and MMD. Admittedly, it's not possible to cover all aspects in one paper, but this would have been a rich addition in my view.

    c. It is mentioned that 49000 out of 50000 samples in the imagenet validation set are used as detection training set. I am curious why use the validation data for this purpose. Why not select 49000 samples from the training set of 1 million+ images? 1000 samples to apply the shift to seems rather small. Can the authors please explain this?

3. Somewhat confusing and overlapping notation. $k$ is used for $\lceil \log_2 m \rceil$ in theorem 4.2 and also as size of window $W_k$. Would improve readability if different variables were used for these two different quantities.

**Questions:**

There are questions of a few technical aspect which I'm hoping the authors can help clarify.

1. It is mentioned that in reference 12, "a risk generalization bound was developed for selective classifiers" but that application of that would require labels, but that theirs doesn't. Given the duality between risk- and coverage-based analysis in the two papers (ref. 12 and this manuscript), can the authors provide some more details to support this statement  please?

2. The bounds are calculated on a dataset of size $m$ while the test window is of a different size $k$. Should these not be matched?

3. What value of $k$ is used for the visualization in Fig. 2? It would be good to understand the impact of the window-size $k$ on the curves for both the no-shift case and the shift case. Specifically, would the coverage bound and empirical coverage continue to remain nearly identical even when $k$ drops to a low value?

4. Should the inequality in Step 9 of Algorithm 1 be $\geq$ instead of $\leq$?

**Limitations:**

No societal impact.

---

> ### Author Rebuttal · Authors · 2023-08-06
>
> Thank you for your feedback! We have addressed each of your concerns below.
>
> **"a. The authors claim that single-instance will not perform as well as their window-based method. However, the only single-instance baselines compared against are softmax response and entropy. These are rather weak baselines and there are several stronger baselines (several of which have been cited by the authors) that should have been compared against."**
>
> In fact, we have tried to address this issue in the last paragraph of the related work section. In general, it is true that including stronger single-instance baselines in our comparison would provide a more robust evaluation. However, the top-performing single-instance methods for out-of-distribution detection require extracting values from each layer in the network [24,25], performing gradient calculations [1], and other costly (time-wise) operations. Therefore, from a practical standpoint, when applied to windows, these methods can significantly grow in complexity, and are non-efficient for the task we consider in this paper; in particular, given the time and space efficiency of our method (see Fig. 1 in the paper). Nevertheless, despite previous works (for example ref. [9,10] in the paper) excluding single-instance methods, we included some.
>
> **"b. It would have been really interesting and useful to see this method applied with different types of uncertainty values apart from softmax (feature-based such as Mahalanobis, predictive uncertainty from Bayesian neural networks, etc.). That would have served as a much stronger evidence of its superiority over other methods such as KS and MMD. Admittedly, it's not possible to cover all aspects in one paper, but this would have been a rich addition in my view."**
>
> We understand your perspective. We Indeed focused on specific uncertainty estimators, and we believe that this serves as a good example to convey the strength of our method. We consider feature-based uncertainty estimator to be beyond the scope of this paper, although we agree that this would be an interesting extension to consider for future work. We'll comment on that in the conclusion section.
>
> **"c. It is mentioned that 49000 out of 50000 samples in the imagenet validation set are used as detection training set. I am curious why use the validation data for this purpose. Why not select 49000 samples from the training set of 1 million+ images? 1000 samples to apply the shift to seems rather small. Can the authors please explain this?"**
>
> Sure, we use the validation data to prevent bias in our detection model. In particular, if we had used data from the training set as the validation data, then the model's uncertainty estimation for this validation data, and the data that is fed to the model during test time, might not be consistent. This inconsistency is equivalent to the difference observed between train data accuracy and test data accuracy. By using a separate validation set, our method ensures that we provide more reliable results.
>
> **"Somewhat confusing and overlapping notation. $k$ is used for $\lceil \log_2 m \rceil$  in theorem 4.2 and also as size of window $W_k$. Would improve readability if different variables were used for these two different quantities."**
>
> Thanks for the note. We will fix this in the final version.
>
> **"It is mentioned that in reference 12, "a risk generalization bound was developed for selective classifiers" but that application of that would require labels, but that theirs doesn't. Given the duality between risk- and coverage-based analysis in the two papers (ref. 12 and this manuscript), can the authors provide some more details to support this statement please?"**
>
> Certainly. The dependence of [12] on labels (their algorithm requires labeled data to operate), basically disqualifies them to be distribution shift detectors, since if labels are provided during test time, there is basically no need to detect a distribution shift. In contrast, our method provides a different approach, which doesn't need labels to operate. We will be happy to further clarify this point in the paper.
>
> **"The bounds are calculated on a dataset of size $m$ while the test window is of a different size $k$. Should these not be matched?"**
>
> No, it is not necessary for the two sample sizes to be matched. This situation is analogous to applying a two-sample t-test, where it is acceptable to have different sample sizes. Therefore, in our case, having different sizes for the in-distribution dataset ($m$) and the test window ($k$) does not pose any issues.
>
> **"What value of $k$ is used for the visualization in Fig. 2? It would be good to understand the impact of the window-size $k$ on the curves for both the no-shift case and the shift case. Specifically, would the coverage bound and empirical coverage continue to remain nearly identical even when $k$ drops to a low value?"**
>
> This is a good point. For Fig. 2 (in the paper) we used a window size of $k=1000$ and indeed as $k$ decreases to lower values ($k=10$), there is less ``agreement'' between the coverage bound and the empirical coverage. To address this point and provide further clarity on this issue, we have prepared additional figures: these are Figures 1a-1h in the attached PDF file. These figures show four more pairs of shift and no-shift cases, for $k=10, k=100, k=200, k=500$; we will incorporate these new figures in an Appendix.
>
> **"Should the inequality in Step 9 of Algorithm 1 be $\leq$ instead of $\geq$ ?"**
>
> After careful review of your comment, we confirm that the inequality should indeed remain $\geq$. Let us explain: let's consider the i-th iteration: if $b_i$ is less than $c^*$, then we'd want to search for larger bounds. Thus, we aim to find smaller $z$ values than the current one (given that $\kappa_f(x)$ is sorted in ascending order). Therefore, to ensure $z$ decreases in the subsequent iteration, we set $z_{max} = z$, so that the use of $\geq$ is correct in this context.

---

> > ### Comment · Reviewer_HyPc · 2023-08-13
> >
> > Thank you to the authors for a detailed response to all reviewer comments. Much appreciated!
> > A few of my concerns have certainly been addressed, in particular the points about
> >
> > + Why the risk generalization bound cannot be applied (since in its formulation as presented in [12], it requires access to labeled data),
> > + Impact of $k$ on the visualization, for which the authors have provided curves, and
> > + The explanation for why dataset size $m$ and window size $k$ can be different (although, I would prefer a more formal justification).
> >
> > A couple of questions/concerns do remain, however:
> >
> > 1. I do think that a lack of benchmarking against stronger single-samples baselines is a weakness though.
> > 2. I am also not convinced by the authors response that they used validation samples to train their detection module because validation data uncertainty estimates might not be consistent with train data uncertainty estimates. That is true, but isn't that always the case for any machine-learning approach? We never train models using samples validation or test set, and there is always a gap between accuracy of a model on training data and accuracy on test data. Why should this paradigm not apply to training the proposed detection module?
> >
> > Just to clarify, I do understand that a separate hold-out validation set of 49000 samples has been created and that final results have not been reported on this held-out set, but on the remaining 1000 samples (which serve as the final test set). However, I still believe that this should have been created from the training data as only then would one be able to assess if the detection module was capable of detecting between true OOD distribution shifts, and not getting confused by train-vs-val shifts.

---

> > > ### Author Response · Authors · 2023-08-14
> > > **Authors response to Reviewer HyPc**
> > >
> > > Thank you for taking the time to read our response in detail. We will address in full below your remaining concerns. We hope that, given our previous response to all your suggestions and comments and the further clarifications below, you would kindly reconsider your rating. Your consideration is highly appreciated!
> > >
> > > **"I do think that a lack of benchmarking against stronger single-samples baselines is a weakness though."**
> > >
> > > As we stated before, we hypothesize that using those stronger single-instance baselines might be impractical in terms of time and space complexity when applied to large windows. Nevertheless, we believe we have a couple of strong arguments for why our method can outperform any suggested single-instance baseline. Let us elaborate.
> > >
> > > Our method leverages an uncertainty estimator, denoted as $\kappa_f$, which serves as its primary engine (refer to Algorithm 2, $\kappa_f$ is an input). Thus, our approach is versatile -- any uncertainty estimator can be integrated with it. In our experiments, we utilized two specific ones: Softmax-response and Entropy. Therefore, we compared our outcomes with the two corresponding single-instance baselines (Softmax-response and Entropy) -- we consistently outperformed these baselines. We want to point out that single-instance baselines are basically using a coverage of 1 (they operate on each sample), and one of the main claims in our paper is that we are *maximizing detection power through lower coverages*, we support this claim in Section 6.2 and with Fig. 2(b) -- which shows that the largest bound deviation is around a coverage of 0.4 (notably less than 1). In particular, for Fig. 2(b) we used target coverages with values\
> > > {$0.1, 0.2, 0.3, 0.4, 0.5, 0.6, 0.7, 0.8, 0.9$} (see lines 219-220 in the paper).
> > > Thus, given any single-instance out-of-distribution detector, we can integrate its uncertainty mechanism ($\kappa_f$) to our method, and by adding a target coverage of $1$ to our list of target coverages values\
> > > ({ 1 } $\cup${0.1, 0.2, 0.3, 0.4, 0.5, 0.6, 0.7, 0.8, 0.9}), we should match or even surpass its detection performance on windows.
> > >
> > > **"I am also not convinced by the authors response that they used validation samples to train their detection module, because validation data uncertainty estimates might not be consistent with train data uncertainty estimates. That is true, but ..."**
> > >
> > > Before diving into the justification for this experimental setting, it's important to highlight that while we could have used the training data to train our detector, doing so would have necessitated simulating the no-shift scenario by drawing 1,000 samples from that same training dataset.
> > > First, we would like to emphasize that, we adopted the common experimental approach, also outlined in references [1,2,24,25], where the no-shift scenario is represented by samples not previously encountered by the model. Thus, we also used a separate validation set.
> > >
> > > The same experimental protocol is also a  standard practice when performing calibration, which is a conceptually similar task whereby you are given a trained model and would like to adjust the model's predictions to better align with the true probabilities of the outcomes.
> > > In the calibration literature, the validation is often called a calibration set. The use of an independent calibration set is indeed required and is a common practice; see for example references [3,4,5].
> > >
> > > To intuitively explain the importance of an independent calibration set imagine you're prepping for an exam using specific questions (the "training set"). If tested on these same questions, you'd excel due to memorization. However, this doesn't mean you'd perform well on new questions (the "test set").
> > > Similarly, in model calibration, training a model on a dataset is like practicing specific questions.
> > > Calibrating with the same data is like being tested on practiced questions, leading to seemingly accurate but potentially misleading results.
> > > On new data, the model might not be as reliable.
> > > In essence, calibrating a model on its training data can give a false sense of its accuracy, risking errors in vital applications.
> > >
> > > We hope this clarifies the point. We remain available to address any further questions or clarifications the reviewer might have.
> > >
> > > **References:**
> > >
> > > [1] - Enhancing the reliability of out-of-distribution image detection in neural networks. ICLR.
> > >
> > > [2] - Generalized ODIN: detecting out-of-distribution image without learning from out-of-distribution data. CVPR.
> > >
> > > [3] - What can we learn from the selective prediction and uncertainty estimation performance of 523 ImageNet classifiers? ICLR.
> > >
> > > [4] - Conformal prediction with missing values. ICML.
> > >
> > > [5] - Conformalized quantile regression. NeurIPS.
> > >
> > > [24] - Detecting out-of-distribution examples with gram matrices. ICML.
> > >
> > > [25] - A simple unified framework for detecting out-of-distribution samples and adversarial attacks. NeurIPS.

---

> > > > ### Comment · Reviewer_HyPc · 2023-08-15
> > > >
> > > > **"it's important to highlight that while we could have used the training data to train our detector, doing so would have necessitated simulating the no-shift scenario by drawing 1,000 samples from that same training dataset"**
> > > >
> > > > I believe this is the key point that we are differing on. I believe that for training the detector, the data samples should be taken from the training split of the dataset of interest, while to calculate metrics such as AUROC, etc. the in-distribution samples should be drawn from the (unperturbed) test split (validation split in case of ImageNet). This is the protocol that is followed in OOD detection work, even in the references listed in your response.
> > > >
> > > > From reference [1]: "At test time, the *test images from CIFAR-10 (CIFAR-100) datasets can be viewed as the in-distribution*
> > > > (positive) examples."
> > > >
> > > > From reference [24]: "For each of these models, we considered the corresponding *test partitions as the in-distribution (positive) examples*."
> > > >
> > > > From reference [25]: "We only use *test datasets* for evaluation."
> > > >
> > > > (emphasis mine in all the above quotes taken directly from the references).
> > > >
> > > > If the in-distribution samples in these papers had been taken from the training set, then their results (AUROC, TPR, etc.) would have been superior compared to what they have reported. Hence, this is a serious issue in my view.
> > > >
> > > > However, I am more than willing to be corrected in case my understanding on the situation is wrong. I look forward to your response.

---

> > > > > ### Author Response · Authors · 2023-08-15
> > > > > **Authors response to Reviewer HyPc, Part 2**
> > > > >
> > > > > After thoroughly reviewing your concerns on this issue, we believe there has been some misunderstanding.
> > > > >
> > > > > Let us start with some definitions,
> > > > > * We are given the partitions $D_{\text{train}}$ and $D_{\text{test}}$ as in ImageNet.
> > > > >
> > > > > To aid the discussion and our experimental objectives, we define these three sets:
> > > > >
> > > > > * $S_{\text{train}}$ : Used for model training (not the detector, the actual neural network).
> > > > > * $S_{\text{val}}$ : Used for the detector's calibration/fitting (specifically for establishing thresholds and bounds in our case).
> > > > > *  $S_{\text{test}}$ : Utilized for evaluating detection performance. In addition to this evaluation dataset, we also use for performance evaluation an OOD (Out-Of-Distribution) dataset. This OOD dataset is not relevant to the present discussion.
> > > > >
> > > > > From our current interpretation of your proposition, the suggested setting is:
> > > > >
> > > > > * $D_{\text{train}} = S_{\text{train}} \cup S_{\text{val}}$.
> > > > > * $D_{\text{test}} = S_{\text{test}}$.
> > > > >
> > > > > In our implementation, the setting is:
> > > > >
> > > > > * $D_{\text{train}} = S_{\text{train}}$.
> > > > > * $D_{\text{test}} = S_{\text{val}} \cup S_{\text{test}}$.
> > > > >
> > > > > Note, in our setting, the random $D_{\text{test}}$ split to $S_{\text{val}} \cup S_{\text{test}}$ has been performed 15 times, to ensure that the set $S_{\text{test}}$ is not consistently the same. **Importantly, in both settings, there is no leakage of information from the test data, $S_{\text{test}}$ (used for performance evaluation) to the training of the model, $S_{\text{train}}$, and the calibration set of the detector, $S_{\text{val}}$.**
> > > > >
> > > > > We claim that these two partition strategies would give the same results. Let us elaborate.
> > > > >
> > > > > Given that both $D_{\text{train}}$ and $D_{\text{test}}$ are sampled i.i.d. from the same distribution $P$, **any partition strategy would produce sets, $S_{\text{train}}$, $S_{\text{val}}$ and $S_{\text{test}}$, which are all sampled i.i.d. from this same distribution $P$**.
> > > > > The only potential concern is the reduced sample size of the test set.
> > > > > In our experiments, the size of $S_{\text{test}}$ is $1,000$ -- the maximum window size.
> > > > > This choice was guided by our empirical findings that most metrics get a nearly perfect score around this size (for the top-performing methods). For example, our method exhibits a score of 99+ for the AUROC, AUPR-In, and AUPR-Out metrics for a window of size $1,000$, for all architectures considered; see Table 2 in the paper.
> > > > >
> > > > > Therefore, since a maximum size of $1,000$ samples seems sufficient for evaluating window-based detection methods, we argue that both settings would yield the same results.
> > > > >
> > > > > Finally, To further clarify our previous points regarding the references mentioned earlier, we have included those key citations:
> > > > >
> > > > > From reference [1]: "The noise magnitude $\epsilon$ was selected on a separate validation dataset, which is different from the out-of-distribution test sets".
> > > > >
> > > > > From reference [2]: "In all experiments, we first train a classifier $f_\theta$ on an in-distribution training set, then tune the hyperparameters (e.g. the perturbation magnitude $\epsilon$) on an in-distribution validation set without using its class labels...".
> > > > >
> > > > > From reference [25]: "we choose the weight of each layer $\alpha_l$ by training a logistic regression detector using validation samples."
> > > > >
> > > > > Thus, our use of a separate validation set is analogous to that in those references.
> > > > >
> > > > > Please let us know if this explanation makes sense to you. We remain available to answer any additional questions or offer further explanations.
> > > > >
> > > > > **References:**
> > > > >
> > > > > [1] - Enhancing the reliability of out-of-distribution image detection in neural networks. ICLR.
> > > > >
> > > > > [2] - Generalized ODIN: detecting out-of-distribution image without learning from out-of-distribution data. CVPR.
> > > > >
> > > > > [25] - A simple unified framework for detecting out-of-distribution samples and adversarial attacks. NeurIPS.

---

> > > > > > ### Comment · Reviewer_HyPc · 2023-08-16
> > > > > >
> > > > > > Let me first express my appreciation to the authors for their willingness to engage in this discussion.
> > > > > >
> > > > > > The way that you defined the problem in terms of $S_{train}, S_{test},$ and $S_{val}$ is precisely the point I was alluding to. Thank you for stating it in crisp and clear terms. Also, I do fully agree that there is no leakage from $S_{test}$ to $S_{train}$ or $S_{val}$. In fact, I stated so as much in one on my previous responses ("*I do understand that a separate hold-out validation set of 49000 samples has been created and that final results have not been reported on this held-out set*").
> > > > > >
> > > > > > The concern was not that there was overlap between test set and val set samples, but rather that a phenomenon similar to the "generalization gap" would occur. This, as the authors are probably well familiar with, is the problem that accuracy on test-set is less than that on train-set even though train and test are sampled i.i.d from the same underlying distribution. This was more my primary concern, and I am still of the opinion that if the protocol followed was $D_{train} = S_{train} \cup S_{val}$ instead of $D_{test} = S_{test} \cup S_{val}$, then there would likely be a drop in performance. Having said that, since the authors mention having validated this across 15 different random splits of $D_{test}$ into $S_{test}$ and $S_{val}$, it does increase confidence in their reported outcomes.
> > > > > >
> > > > > > The authors are also right that the stated references have used validation sets to obtain values of hyper-parameters like $\epsilon$ and $\alpha_l$. This, I feel, is more in the spirit of hyper-parameter tuning and would be akin to the authors using a validation set to determine the optimal value of window-size $k$, but not to train the detection module itself. Nonetheless, the other methods are in fact using small held-out validation sets sampled from the test; so thanks for pointing this out.

---

### Official Review · Reviewer_MZpo · 2023-07-21

**Soundness:** 3 good
**Presentation:** 2 fair
**Contribution:** 3 good
**Rating:** 6
**Confidence:** 3

**Summary:**

The authors tackle the problem of out-of-distribution detection on a stream of instances. They introduce a method for coverage-based detection (CBD) that creates bounds and thresholds for monitoring by applying a newly derived selective with guaranteed coverage algorithm and then conducts a t-test on the violation sum between a window of newly received samples and a given detection dataset. The method is then analyzed for complexity and evaluated experimentally against ImageNet with various architectures. The method scales up with the size of the detection-training set size given more favourable complexity than previous baselines.

**Strengths:**

1. The case is made that at large scales of data the provided method is more practical for implementation and can match or outperform existing methods for detecting distribution shift on streams of data.

2. The authors provide a wide range of data and analysis to back up their claims:
2.1 Derivation of the SGC and CBD algorithms for estimating coverage and detecting violations.
2.2 Lower complexity
2.3 Experimental validation on ImageNet with various architectures and types of distribution shifts.
2.4 Showing a walkthrough of bound and threshold identification which helps

3. They provide a thorough explanation of their method and how it differs from existing baselines like KS and MMD based tests that are popular both in research and industry. Based on the results here I could easily see CBD replacing either of these methods in existing setups.


**Weaknesses:**


1. Experimental setup:Window-based methods in any task rely quite heavily on the dataset in question. It would have been great to see a more diverse set of benchmarks for out-of-distribution detection if they could be modified to the given problem statement. E.g. Wilds (Koh et. al., 2021) That being said I appreciate the various shifts applied to test on different kinds of distribution shifts.

2. Presentation: I found Table 2 hard to parse and dense. I understand it is hard to fit in all the data but there could be some improvements.

3.

**Questions:**

1. It seems that while the method scales with the size of the detection-training set, it converges with other methods as the window size increases (except with ViT-T architecture). Increasing the window size leads to a bigger jump in evaluations like Detection Error once the window size is above a certain threshold. What is the relationship between m, k with respect to out-of-distribution evaluation and is there any reason why someone might prefer to scale up the window size rather than the detection-training set and thus nullify the complexity advantages of this method?

**Limitations:**

1. Requiring access to the final layer of weights in the model rather than being fully black-box like some of the other methods.
2. The authors don't highlight any particular limitations for practitioners to watch out for. Even suggestions around tuning the threshold parameters in different data settings could be helpful (or perhaps they are correct that the defaults they use are sufficient for all cases).

---

> ### Author Rebuttal · Authors · 2023-08-07
>
> We appreciate your positive feedback and suggestions. We are glad that you believe our method could serve as a superior alternative to the baselines in distribution shift detection tasks! Below, we will address each of the points you raised.
>
> **"Experimental setup:Window-based methods in any task rely quite heavily on the dataset in question. It would have been great to see a more diverse set of benchmarks for out-of-distribution detection if they could be modified to the given problem statement. E.g. Wilds (Koh et. al., 2021) That being said I appreciate the various shifts applied to test on different kinds of distribution shifts."**
>
> We agree that the application of diverse OOD datasets, like those in Wilds (Koh et. al., 2021) could enhance our work's comprehensiveness. However, our methodology was inspired by ref. [10] in the paper, and expanded upon it, both in terms of the OOD datasets employed and the variety of models utilized.
>
> **"Presentation: I found Table 2 hard to parse and dense. I understand it is hard to fit in all the data but there could be some improvements."**
>
> Indeed it is due to space constraints. we will do our best to improve its readability in the final version of the paper.
>
> **"It seems that while the method scales with the size of the detection-training set, it converges with other methods as the window size increases (except with ViT-T architecture). Increasing the window size leads to a bigger jump in evaluations like Detection Error once the window size is above a certain threshold. What is the relationship between m, k with respect to out-of-distribution evaluation and is there any reason why someone might prefer to scale up the window size rather than the detection-training set and thus nullify the complexity advantages of this method?"**
>
> Regarding $m$ (the detection-training set size), the ideal strategy would be to maximize it as much as possible. The reason is that, as $m$ increases, a detector captures the underlying distribution of the in-distribution (ID) data more accurately. In particular, for ``Google-Scale'' datasets (approximately a billion images), the reliability of a detector is questionable without fitting it to a sufficiently large $m$.
> Regarding the window size $k$, its selection depends on the user's specific application. However, we wish to emphasize, that the top-performing methods achieve a nearly perfect score in all metrics for sufficiently large $k$, for example, our method exhibits a score of 99+ for the metrics AUROC, AUPR-In, and AUPR-Out for $k=1000$, for all architectures considered; see Table 2 in the paper. Thus, The intriguing question, in our opinion, is which method excels for smaller $k$ values.
>
> **"Requiring access to the final layer of weights in the model rather than being fully black-box like some of the other methods."**
>
> We would like to clarify that neither our method nor the baseline methods needs access to the final layer of weights. Our approach employs only the softmax function, while the baseline methods utilize either the softmax or the embedding layer, depending on the specific baseline in question.
>
> **"The authors don't highlight any particular limitations for practitioners to watch out for. Even suggestions around tuning the threshold parameters in different data settings could be helpful (or perhaps they are correct that the defaults they use are sufficient for all cases)."**
>
> We refer the reviewer to the second paragraph of the introduction section, specifically lines 33-41, where we discuss the limitations of our work. To clarify, our study's primary limitation is that we don't consider identifying single-instance out-of-distribution examples, characterizing types of distribution shifts, or addressing the complementary topic of out-of-distribution robustness.

---

> > ### Comment · Reviewer_MZpo · 2023-08-14
> >
> > Thanks for your detailed response.
> >
> > After further reading through [10] and some of the more recently published works which cite it there are clearly a range of further evaluation methods that have been proposed. Maggio & Dreyfus-Schmidt in their 2021 work "Ensembling Shift Detectors: an Extensive Empirical Evaluation" show synthetic shifts applied to 21 different classification tasks from OpenML and Kaggle. This seems like a more general evaluation than the currently proposed one which would improve the ability to evaluate the methodology. However, the methods proposed in this work are sound enough that I will keep my current rating without any additional empirical evaluation.
> >
> > As for the other questions, your responses are satisfying.

---

### Official Review · Reviewer_a2HS · 2023-07-25

**Soundness:** 3 good
**Presentation:** 3 good
**Contribution:** 3 good
**Rating:** 7
**Confidence:** 2

**Summary:**

This paper draws attention to the problem of window-based distribution shift detection and introduces a new method for it. The method is theoretically sound and demonstrated empirically to be effective on datasets with multiple types of shifts.

**Strengths:**

- The proposed method is evaluated on extensive datasets with various types of shifts and outperforms selected baselines.
- The computational complexity is impressively low.
- Due to its lower computational complexity, the proposed method is scalable compared to the baselines.

**Weaknesses:**

There aren't significant weaknesses in my opinion, but several minor ones might need clarification:
- I am concerned with the novelty of Algorithm 1, selection with guaranteed coverage, since most parts here are similar to SGRC in [1].  Is coverage the only modification here?
- The paper set out with a window-based shift detection task, which feels like a generalization from the single-instance task to me. However, there are no results of $|W_k|=1$. Can the proposed method still outperform baselines in this case? (Authors don't have to add new experiments here).
- The goal is to detect if there is a shift in a window $W_k$ of test data. I'd like to know how the proposed method will behave when only a portion of $W_k$ is from a different distribution.

[1] "Selective Classification for Deep Neural Networks", 2017

**Questions:**

- In equation 2, what does $|m\cdot \hat c(\theta, S_m)|$ intuitively stand for? Why not just add up to $m$?
- Is this typo? line 55: "read-world".

**Limitations:**

See the weaknesses above.

---

> ### Author Rebuttal · Authors · 2023-08-06
>
> We appreciate your positive feedback! We are glad you have found our approach to be effective and scalable. Below, we reply to more specific comments and questions.
>
> **"I am concerned with the novelty of Algorithm 1, selection with guaranteed coverage, since most parts here are similar to SGRC in [1]. Is coverage the only modification here?"**
>
> We respect your comment. In relation to paper [1], coverage, rather than risk, is a major innovation of Algorithm 1. However, we wish to emphasize that SGRC, as outlined in [1], fundamentally relies on labeled data for its operation, which disqualifies it as an engine for detecting distribution shifts, where labels are unavailable. Thus, although Algorithm 1 and SGRC are syntactically similar, they are semantically completely different in the sense that they are designed for completely different use cases. By focusing on coverage, our technique offers a unique feature that isn't found in [1].
>
> **"The paper set out with a window-based shift detection task, which feels like a generalization from the single-instance task to me. However, there are no results of $W_k=1$. Can the proposed method still outperform baselines in this case? (Authors don't have to add new experiments here)."**
>
> We would like to clarify that our method (nor any of the population-based baselines) supports a window size of $W_k=1$. This is primarily due to the fact that coverage is a property based on population. If we attempt to apply our method to a single sample only, the empirical coverage would be binary -- covered (signifying an empirical coverage of 1), or not covered (yielding an empirical coverage of 0).
>
> **"The goal is to detect if there is a shift in a window of test data. I'd like to know how the proposed method will behave when only a portion of $W_k$ is from a different distribution."**
>
> Following your suggestion, we attach here results from a preliminary experiment under a nearly identical setting, which we ran in the early stages of our research and wasn't included in the paper. In this experiment, we deliberately adjusted the proportion ratio $p$ to allow a mix of OOD and ID samples within a window. The ImageNet dataset was used as in-distribution (ID) and EfficientNetB0 was chosen as the model. In Table 3 of the attached PDF file, we present results for a proportion ratio of $p=1/3$ (i.e., one-third of the window is contaminated).
>
> We will be happy to include (and extend -- should time allow) these preliminary results in the final version of the paper if the reviewer thinks that this is important!
>
> **"In equation 2, what does $|m\cdot \hat{c}(\theta, S_m) |$ intuitively stand for? Why not just add up to $m$?"**
>
> Let us clarify: $|m\cdot \hat{c}(\theta, S_m) |$ stands for the number of samples covered if using the threshold $\theta$; see the equation of empirical coverage, $\hat{c}(\theta, S_m)$ (in line 186), and the equation for $g_\theta(x)$ (in line 183). We wish to point out, that adding up to $m$ will correspond to an empirical coverage of 1, and our method relies on the idea of using coverages lower than 1. In fact, in Section 6.2 we demonstrate the effectiveness of using coverages lower than 1. In particular, in Fig. 2 we demonstrate the bound violation by the red circles (which indicate a distribution shift), as a function of the target coverage; the largest violation occurs around a target coverage of 0.4.
>
> **"Is this typo? line 55: "read-world"."**
>
> Thanks for this note. This is indeed a typo, and we will fix it to ``real-world''.

---

> > ### Comment · Reviewer_a2HS · 2023-08-12
> >
> > Thanks for the authors' responses. After reading comments from other reviewers, I agreed on several points:
> >
> > - Reviewer **7FwS**'s concern about the limited performance gap with large window size. However, it was noted by authors that detection with a smaller window size implies a "harder" scenario ("The intriguing question, in our opinion, is which method excels for smaller $k$ values"). And also, in my opinion, the scalability of the proposed method could excuse away the narrowing performance gap.
> > - Reviewer **HyPc**'s concern about window size value in Fig 2. The additional visualisation pairs provided in the attachment show consistency with Fig 2.
> > - Authors' comment on my second concern and Reviewer **M7ue**'s first concern help understand the motivation behind the proposed method.
> >
> > Overall, the authors have addressed most of my concerns. Thus I would like to keep my original score to support the acceptance.

---

### Author Rebuttal · Authors · 2023-08-07

We are grateful to all reviewers for their feedback and constructive comments and happy to see that in general our work was positively received (4 reviewers with a score  $\geq$ 6).

In particular, the validity and importance of our proposed method have been specifically recognized:
* "attractive problem" (**BTJN**)
* "The idea of detecting OOD samples within a window is interesting" (**M7ue**)
* "Based on the results here I could easily see CBD replacing either of these methods in existing setups" (**MZpo**)
* "a very important problem with multiple applications" (**7FwS**)
* "The case is made that at large scales of data the provided method is more practical" (**MZpo**)

As an important part of our work, we have proposed a significant improvement in detection complexity. This aspect has been positively appreciated as well:

* "The computational complexity is impressively low" (**a2HS**)
* "more favorable complexity than previous baselines" (**MZpo**)
* "The algorithm exhibits great time complexity, space complexity, and good performance" (**UhJJ**)
* "The results demonstrate the method is faster and requires less memory than the current SoTA" (**7FwS**)
* "Due to its lower computational complexity, the proposed method is scalable compared to the baselines" (**a2HS**)

Finally, we were also very happy to notice that the reviewers have found our experimental setting to be "comprehensive and well-designed" (**7FwS**), `"The proposed method is evaluated on extensive datasets with various types of shifts and outperforms selected baselines" (**a2HS**), and that the paper is "well written" (**BTJN**, **7FwS**), "easy to follow" (**UhJJ**), and "well motivated" (**HyPc**).

For the final version of our paper, we will take into consideration all the comments, suggestions, and other feedback provided by the reviewers. In particular, as requested by **BTJN**, we present new results over the CW attack as the out-of-distribution (OOD) dataset, see Table 2 in the attached PDF file. Furthermore, reviewers **M7ue**, **UhJJ** suggest that an experiment with windows containing a mix of both in-distribution and out-of-distribution samples will strengthen the paper. Thus, we also present a preliminary experiment along this line in the attached PDF file, see Table 3. In addition, to the specific request of reviewer **HyPc**, we show the new Figure 1 in the attached PDF file, which effectively illustrates the shift/no-shift scenarios for values of $k$ other than the specified value of $k=1000$.
Finally, in response to reviewers **M7ue** and **7FwS** requests to verify the generalizability of our method to other in-distribution datasets, we've also included preliminary results that confirm this issue. These results, which use CIFAR10 as the in-distribution dataset, can be found in Table 1 of the attached PDF file.

Next, we provide responses to each reviewer so as to address their comments in detail and answer their specific questions.

---

### Decision · Program_Chairs · 2023-09-21

**Decision:**

Accept (poster)

**Comment:**

All reviewers voted for accept. After discussion, we agree that this work merits a NeurIPS acceptance.